# LexEval: A Comprehensive Chinese Legal Benchmark for Evaluating Large Language Models

**Haitao Li**
Department of Computer Science
Tsinghua University
Beijing, China 100000
liht22@mails.tsinghua.edu.cn

**You Chen**
Department of Computer Science
Tsinghua University
Beijing, China 100000
chenyou21@mails.tsinghua.edu.cn

**Qingyao Ai** [*]
Quan Cheng Laboratory
Department of Computer Science
Tsinghua University
Beijing, China 100000
aiqy@tsinghua.edu.cn

**Yueyue Wu**
Quan Cheng Laboratory
Department of Computer Science
Tsinghua University
Beijing, China 100000
wuyueyue@mail.tsinghua.edu.cn

**Ruizhe Zhang**
Department of Computer Science
Tsinghua University
Beijing, China 100000
u@thusaac.com

**Yiqun Liu**
Quan Cheng Laboratory
Department of Computer Science
Tsinghua University
Beijing, China 100000
yiqunliu@tsinghua.edu.cn

## Abstract

Large language models (LLMs) have made significant progress in natural language processing tasks and demonstrate considerable potential in the legal domain. However, legal applications demand high standards of accuracy, reliability, and fairness. Applying existing LLMs to legal systems without careful evaluation of their potential and limitations could pose significant risks in legal practice. To this end, we introduce a standardized comprehensive Chinese legal benchmark LexEval. This benchmark is notable in the following three aspects: (1) Ability Modeling: We propose a new taxonomy of legal cognitive abilities to organize different tasks. (2) Scale: To our knowledge, LexEval is currently the largest Chinese legal evaluation dataset, comprising 23 tasks and 14,150 questions. (3) Data: we utilize formatted existing datasets, exam datasets and newly annotated datasets by legal experts to comprehensively evaluate the various capabilities of LLMs. LexEval not only focuses on the ability of LLMs to apply fundamental legal knowledge but also dedicates efforts to examining the ethical issues involved in their application. We evaluated 38 open-source and commercial LLMs and obtained some interesting findings. The experiments and findings offer valuable insights into the challenges and potential solutions for developing Chinese legal systems and LLM evaluation pipelines. The LexEval dataset and leaderboard are publicly available at `https://github.com/CSHaitao/LexEval` and will be continuously updated.

---

[*]Corresponding author.

38th Conference on Neural Information Processing Systems (NeurIPS 2024) Track on Datasets and Benchmarks.

# 1 Introduction

Recently, the rapid development of large language models (LLMs) has brought new opportunities to the research of general artificial intelligence. A series of models (e.g., ChatGPT), with their extensive knowledge and outstanding language processing ability, have demonstrated excellent performance in various language processing tasks such as text generation, machine translation, and dialogue systems [10, 4, 6, 45, 37]. Meanwhile, LLMs have profoundly impacted the work patterns of legal practitioners and the development of the legal field. Recent studies show that the GPT-4 has the ability to pass the U.S. Judicial Exam [23]. By interacting with large language models, lawyers and judges can analyze legal documents more efficiently, obtaining comprehensive and valuable information and judicial advice. This has led to a growing trend among legal practitioners to incorporate LLMs as a vital supportive instrument in legal proceedings[11, 25, 27].

Despite the considerable potential of large language models, there are still concerns about their application in the legal domain [38, 35, 26]. Firstly, unlike human decision-making, which is grounded in professional knowledge and logical reasoning, LLMs derive decisions from patterns and connections extracted from massive amounts of training data. Consequently, these models, predicated on probabilistic frameworks, often fall short of ensuring the reliability and explainability of their output [18]. Additionally, existing research has indicated that LLMs may produce misleading and factually incorrect content [34]. Substandard legal texts or flawed judicial guidance may mislead legal practitioners and increase their workload. Finally, the content generated by LLMs may reflect biases present in the training data, leading to unfair treatment of certain groups or specific events. This may undermine the effectiveness and fairness of judicial proceedings and judgments, bringing considerable systemic risks.

The great potential and inherent risks of LLMs in the legal domain give rise to the urgent need for a standardized and comprehensive benchmark [41, 48]. Such a benchmark is essential to ensure that LLMs meet the high standards required for legal practice, minimizing the risks while maximizing their beneficial impact. Although numerous methods for evaluating the abilities of LLMs have been developed, most focus on assessing their generalist abilities on non-professional or semi-professional texts. These benchmarks provide limited guidance for highly specialized fields such as the legal domain[54, 22, 5]. For instance, the well-known Chinese language model evaluation framework, C-Eval [22], primarily uses test questions from high school and university courses. However, in judicial applications, tasks like case summarization, legal case retrieval, and judgment prediction require LLMs to consider precise legal knowledge and complex legal contexts. These tasks often involves highly specialized elements such as judicial interpretation and reasoning. To the best of our knowledge, existing general evaluation benchmarks are unable to reflect or capture the complexity of judicial cognition and decision-making. Furthermore, some researchers have utilized existing traditional natural language processing datasets to construct benchmarks, such as LawBench [17] and LaiW [13], to evaluate the performance of LLMs in the Chinese legal system. However, traditional datasets are typically designed to test specific capabilities from a computer-centric perspective, which does not always reflect the practical use of LLMs in legal applications. Moreover, these benchmarks often overlook aspects such as legal ethics, which are crucial for ensuring the safe application of LLMs in the legal domain. Also, the evaluation metrics for previous tasks vary significantly, complicating the standardization of model performance measurement. Simply integrating existing datasets cannot provide a standardized and comprehensive evaluation of LLMs' capabilities in the legal domain.

In light of these limitations, we present LexEval: a comprehensive Chinese legal benchmark for evaluating LLMs. LexEval focuses on practical legal applications, involving how legal professionals manage, contemplate, and resolve legal issues. Firstly, to systematically organize various evaluation tasks, we propose a Legal Cognitive Ability Taxonomy (LexAbility Taxonomy), which includes six aspects: Memorization, Understanding, Logic Inference, Discrimination, Generation, and Ethic. This taxonomy comprehensively analyzes various legal tasks and their intrinsic connections, constructing a systematic framework for evaluating LLMs. Then, based on the LexAbility Taxonomy, we collect 14,150 questions covering 23 legal tasks. To our knowledge, LexEval is the largest and most comprehensive Chinese legal benchmarking dataset for evaluating LLMs. Moreover, LexEval is constructed from existing legal datasets (reorganized into a unified format), exam datasets in reality, and manually curated datasets. It adopts standard evaluation methods and metrics, laying a solid groundwork for future expansion and the integration of diverse tasks. It's important to acknowledge that despite its comprehensiveness, LexEval may not cover every practical application task within the

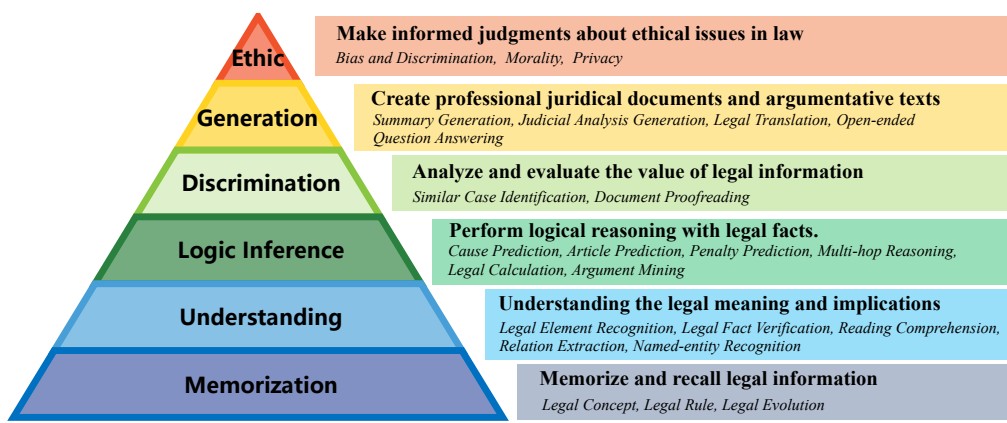

Figure 1: Overview of the legal cognitive ability taxonomy.

legal field. As a platform supporting further research, LexEval encourages individuals to contribute additional tasks to the taxonomy, collectively pushing the boundaries of what's achievable in the field of legal language understanding and generation. We conduct a thorough evaluation of 38 popular LLMs, including General LLMs and Legal-specific LLMs. The experimental results show that the existing LLMs are ineffective and unreliable in addressing legal problems. We hope this benchmark can point out different directions for future work.

## 2 Related Work

In recent years, large language models (LLMs) have drawn great attention in academia and industry for their excellent performance and wide applicability [36, 50, 28]. Models such as ChatGPT and ChatGLM achieve excellent performance across various tasks through mechanisms such as pre-training, supervised fine-tuning, and alignment with human or AI feedback [2, 8, 15, 16]. By learning from massive amounts of text data, LLMs can capture the subtle differences and complex patterns of language, demonstrating the great potential in understanding and generating human language.

However, despite great success, they face significant challenges in the legal domain [32, 7, 14, 31]. In the legal domain, accuracy, reliability, and fairness are crucial, but LLMs often perform poorly in these aspects due to issues like hallucination [32] and inherent biases [52, 9, 29]. Hallucination refers to models generating information that is not based on facts, which can lead to misleading or entirely incorrect conclusions in legal documents and consultations. Additionally, due to biases in the training data, the model may inadvertently replicate and amplify these biases, affecting its fairness and accuracy in applications such as legal judgment prediction, case analysis, and contract review.

To mitigate these issues, the community has proposed a series of evaluation criteria and benchmarks [19, 17, 13, 30]. For example, LegalBench [19] is dedicated to the collaborative evaluation of legal reasoning tasks in English LLMs, consisting of 162 tasks contributed by 40 contributors. Lawbench [17] and LaiW [13] have conducted evaluations on the Chinese legal system using existing traditional natural language processing datasets, contributing to the development of the community. However, these datasets all focus on the partial performance of LLMs and do not provide a comprehensive evaluation. In this paper, we devote to a more comprehensive evaluation of the performance of LLMs in the legal domain. Leveraging the proposed legal cognitive ability taxonomy, we constructed the largest legal benchmark in the Chinese community through various means.

## 3 LexEval

### 3.1 Design Principle

The motivation behind LexEval is to help developers quickly understand the capabilities of LLMs within the legal domain across multiple dimensions, enabling them to focus on specific areas for enhancement. To this end, we advocate for considering the hierarchy and connections of abilities, rather than organizing evaluations based solely on difficulty or in a discrete manner. Nevertheless,

research on the hierarchical abilities of LLMs is still in the early stages, and to our knowledge, there isn't a well-developed taxonomy describing the abilities of LLMs in legal applications[39]. Drawing inspiration from Bloom's taxonomy [24] and real-world legal application scenarios, we propose a legal cognitive ability taxonomy (LexAbility Taxonomy) to guide the organization of tasks in LexEval.

As depicted in Figure 1, the taxonomy categorizes the application of LLMs in the legal domain into six ability levels: Memorization, Understanding, Logic Inference, Discrimination, Generation, and Ethic. At the Memorization level, LLMs are tasked with memorizing and recalling legal information, including fundamental legal statutes, case law, basic legal principles, and specialized legal terminology, among other essential content. Moving to the Understanding level, LLMs must demonstrate an aptitude for comprehending the meaning and implications of legal information. They should possess the ability to interpret legal concepts, texts, and issues accurately. Logical Inference involves the capacity for legal reasoning and deductive logic. LLMs should be capable of deducing conclusions based on provided legal facts and rules, identifying and applying legal patterns and principles effectively. The Discrimination level necessitates LLMs to analyze and evaluate the significance of legal information according to specific criteria. At the Generation level, LLMs are expected to produce professional legal documents and argumentative texts within specific legal contexts. This includes drafting legal writings, contracts, and providing legal opinions. LLMs should generate precise, legally sound, and well-structured texts based on given conditions and requirements. Finally, the Ethics level requires LLMs to make judgments about ethical issues in the legal domain. Models should identify and analyze legal ethical issues, make ethical decisions, and weigh advantages and disadvantages. They must consider ethical principles of law, professional ethics, and social values in their decision-making processes.

Each level contains several specific evaluation tasks corresponding to the respective abilities. Legal practitioners can employ this taxonomy to identify the cognitive levels attained by LLMs, thereby enhancing the planning of training objectives and downstream applications. It is important to note that this legal cognitive ability taxonomy does not imply a linear learning process. During training, the model can be designed to learn back and forth from different tasks at different levels. Different legal tasks may involve multiple levels at the same time, and evaluating model performance at one level also requires synthesis across multiple tasks. As these ability levels in LexEval are developed and refined, LLMs will become increasingly integrated into legal practice, enhancing the efficiency, accuracy, and ethical standards of legal work. This taxonomy not only provides a framework for assessing the current capabilities of LLMs but also guides future advancements in the field. Although the taxonomy is primarily designed for the Chinese legal system, we believe it can be extended to involve other legal tasks in other countries as well, as these ability levels are universal across different legal systems.

## 3.2 Data Collection and Processing

**Data Source:** The data in LexEval comes from three sources. The first source comprises existing datasets and corpora, primarily including CAIL [2], JEC-QA [53], and LeCaRD [33]. As these resources are originally designed for non-LLM evaluation settings, we standardize the data format and adjust the prediction targets to align with the evaluation objectives of LLMs. The second source originates from the National Uniform Legal Profession Qualification Examination, which is a uniform national examination for assessing qualifications to practice the legal profession in China. We carefully select and adapt exam questions from previous years to suit our evaluation framework. The third task source is expert annotation, where we hire 18 experts in the legal field as annotators to craft precise and relevant evaluation questions. Detailed data sources and licenses can be found in Appendix C.

**Data Processing:** We collect data in various formats, including PDF, JSON, LaTeX, and Word, among others. By using techniques such as OCR, we first convert PDF and Word documents into textual form. For those questions that are difficult to parse automatically, we process them manually. Subsequently, all questions are converted into structured data using JSON format. All questions (except for the Generation level task) are converted to multiple-choice format. This is because multiple-choice questions have clearly defined metrics (i.e., accuracy) and are also a simple and effective way to evaluate the capabilities of the LLMs, which have been widely used in various benchmarks [17, 13, 22]. Detailed construction processes for each task can be found in Appendix

---

[2]The data from CAIL can be found on the official website of the competition http://cail.cipsc.org.cn/.

C.3. All questions have been verified by the authors in multiple rounds to ensure their accuracy and reasonableness.

**Data Annotation:** For tasks lacking existing datasets, we hire professional annotators to create entirely new datasets. Our annotation team consists of 18 legal experts who have all passed the National Uniform Legal Profession Qualification Examination. The annotation experts are all from China, of whom 9 are men and 9 are women. Before the beginning of the annotation work, we signed a legally effective agreement with all annotation experts to protect their rights and interests. To ensure the quality of annotation, all annotators first go through several hours of interpretation to understand their respective tasks. After that, we provide several examples to help them understand the format of tasks. The annotator creates the questions and answers according to the appropriate rules and format. Our gold annotators, who hold a Ph.D. in law, cross-check and inspect all generated questions. Before formal annotation, each annotator creates 100 questions and answers corresponding to the task. Subsequently, only annotators who achieve a 90% approved rate through cross-checking and inspection are allowed to annotate formally. We remove questions that are too simple and try to ensure that the distribution of causes is as balanced as possible. For each approved question, we pay the legal expert 0.7 dollars. We have annotated a total of 6,250 questions, with a total payment of 4,375 dollars. Detailed annotation guidance for each task can be found in Appendix C.3 and Appendix F.

Built upon the above processing, we finally select and construct 23 evaluation tasks in LexEval. For the existing datasets, we try our best to avoid using datasets that have already been extensively mined by existing LLMs (e.g. C-Eval) so that the risk of test data leakage could be minimized. To ensure the quality of LexEval, we also try to balance the distributions of legal documents from different causes, thereby avoiding bias or long-tail effects in the dataset.

## 3.3 Task Definition

Based on the legal cognitive ability taxonomy, we construct a series of evaluation tasks. Table 1 shows the overview of tasks in LexEval. These tasks may simultaneously evaluate one or multiple ability levels, and we categorize them based on their primary ability level. Each task has at least 100 evaluation samples. Among these tasks, 11 tasks are derived from existing datasets, 2 tasks come from the National Uniform Legal Profession Qualification Examination, and 10 tasks are annotated by experts. Detailed task definition, construction process, and task statistics can be found in Appendix C.4. Based on these tasks, LexEval not only provides comprehensive coverage of legal knowledge and reasoning ability but also detects issues such as bias and discrimination in legal ethics, providing valuable insights for in-depth evaluation and analysis.

## 3.4 Legal and Ethical Considerations

Due to the sensitivity of the legal domain, we conducted a thorough review for this benchmark. All open-source datasets we utilized are licensed. LexEval tasks are subject to different licenses. Appendix C.1 provides a summary of the licenses. The authors take full responsibility for any infringement and confirm the authorization of the dataset. Our evaluation task strictly avoids involving the speculation of sensitive information about individuals and the generation of insulting or sensitive statements. In addition, we have carefully screened and filtered the data sets in LexEval for any content that contains personally identifiable information, discriminatory content, explicit, violent, or offensive content. The data set has been ethically reviewed by legal experts. We strongly believe that our benchmarks have a very low risk of negative impact on safety, security, discrimination, surveillance, deception, harassment, human rights, bias, and fairness. Appendix B.2 discusses the potential social impacts.

# 4 Evaluation

In this section, we present the experimental setup, evaluated models, and experimental results.

Table 1: Details of tasks within LexEval.

| Level | ID | Task | Metrics | Data Source | Test Set |
|---|---|---|---|---|---|
| Memorization | 1-1 | Legal Concept | Accuracy | JEC-QA [53] | 500 |
| | 1-2 | Legal Rule | Accuracy | Expert Annotation | 1000 |
| | 1-3 | Legal Evolution | Accuracy | Expert Annotation | 300 |
| Understanding | 2-1 | Legal Element Recognition | Accuracy | CAIL-2019 | 500 |
| | 2-2 | Legal Fact Verification | Accuracy | Expert Annotation | 300 |
| | 2-3 | Reading Comprehension | Accuracy | CAIL-2021 | 100 |
| | 2-4 | Relation Extraction | Accuracy | CAIL-2022 | 500 |
| | 2-5 | Named-entity Recognition | Accuracy | CAIL-2021 | 500 |
| Logic Inference | 3-1 | Cause Prediction | Accuracy | CAIL-2018 | 1000 |
| | 3-2 | Article Prediction | Accuracy | CAIL-2018 | 1000 |
| | 3-3 | Penalty Prediction | Accuracy | CAIL-2018 | 1000 |
| | 3-4 | Multi-hop Reasoning | Accuracy | Exams | 500 |
| | 3-5 | Legal Calculation | Accuracy | Expert Annotation | 400 |
| | 3-6 | Argument Mining | Accuracy | CAIL-2021 | 500 |
| Discrimination | 4-1 | Similar Case Identification | Accuracy | LeCaRD [33]&CAIL-2019 | 500 |
| | 4-2 | Document Proofreading | Accuracy | Expert Annotation | 300 |
| Generation | 5-1 | Summary Generation | Rouge-L | CAIL-2020 | 1000 |
| | 5-2 | Judicial Analysis Generation | Rouge-L | Expert Annotation | 1000 |
| | 5-3 | Legal Translation | Rouge-L | Expert Annotation | 250 |
| | 5-4 | Open-ended Question Answering | Rouge-L | Exams | 500 |
| Ethic | 6-1 | Bias and Discrimination | Accuracy | Expert Annotation | 1000 |
| | 6-2 | Morality | Accuracy | Expert Annotation | 1000 |
| | 6-3 | Privacy | Accuracy | Expert Annotation | 500 |

## 4.1 Setup

We evaluate the LLMs in both zero-shot and few-shot settings. In the zero-shot setting, the inputs to LLMs are only instructions and queries. In the few-shot setting, we design three different examples for each task. These examples can be found on the GitHub website. When evaluating LLMs, we set the temperature to 0 to minimize the variance introduced by random sampling. For chat LLMs, we reserve the format of their dialog prompts. When the input length exceeds the maximum context length of LLMs, we truncate the input sequence from the middle since the front and end of the input may contain crucial information. The input prompts used during our evaluation can be found in the Appendix C.4. We standardize our evaluation metrics by using Accuracy to evaluate all multiple-choice questions and Rough-L to evaluate tasks at Generation level.

The evaluation metrics for each task can be found in Table 1. We also discuss the limitations of the evaluation metrics in Appendix B.1.

## 4.2 Evaluated Models

We evaluate a total of 38 popular models, categorized into two main groups: General LLMs and Legal-specific LLMs.

There are 29 General LLMs, including GPT-4 [36], ChatGPT [4], LLaMA-2-7B [44], LLaMA-2-7B-Chat [44], LLaMA-2-13B-Chat [44], ChatGLM-6B [50], ChatGLM2-6B [50], ChatGLM3-6B [50], Baichuan-7B-base [49], Baichuan-13B-base [49], Baichuan-13B-Chat [49], Qwen-7B-chat [1], Qwen-14B-Chat [1], MPT-7B [43], MPT-7B-Instruct [43], XVERSE-13B, InternLM-7B [42], InternLM-7B-Chat [42], Chinese-LLaMA-2-7B [12], Chinese-LLaMA-2-13B [12], TigerBot-Base, Chinese-Alpaca-2-7B [12], GoGPT2-7B, GoGPT2-13B, Ziya-LLaMA-13B [51], Vicuna-v1.3-7B, BELLE-LLAMA-2-13B [3], Alpaca-v1.0-7B, MoSS-Moon-sft [40].

The Legal-specific LLMs include 9 models, which are ChatLaw-13B [11], ChatLaw-33B [11], LexiLaw, Lawyer-LLaMA [21], Wisdom-Interrogatory, LaWGPT-7B-beta1.0, LaWGPT-7B-beta1.1, HanFei [20], Fuzi-Mingcha [46].. The specific description of evaluated models can be found in the Appendix D.

Table 2: Zero-shot performance(%) of various models at Memorization, Understanding, and Logic Inference level. Best preformance in each column is marked bold.

| Model | Memorization(Acc.) | | | Understanding(Acc.) | | | | | Logic Inference(Acc.) | | | | | |
|---|---|---|---|---|---|---|---|---|---|---|---|---|---|---|
| | 1-1 | 1-2 | 1-3 | 2-1 | 2-2 | 2-3 | 2-4 | 2-5 | 3-1 | 3-2 | 3-3 | 3-4 | 3-5 | 3-6 |
| GPT-4 | **34.0** | 35.4 | 14.0 | 79.8 | **51.0** | **94.0** | 78.0 | **96.2** | **80.3** | 68.3 | **53.7** | **33.2** | **66.0** | **57.8** |
| Qwen-14B-Chat | 28.0 | 38.6 | 11.4 | **93.4** | 45.3 | 90.0 | **85.6** | 91.8 | 80.2 | **91.0** | 27.9 | 31.6 | 44.7 | 50.4 |
| Qwen-7B-Chat | 22.8 | **38.9** | 8.4 | 79.8 | 43.3 | 87.0 | 67.2 | 92.0 | 79.2 | 83.9 | 53.2 | 24.2 | 36.3 | 45.0 |
| ChatGPT | 19.0 | 25.6 | 9.0 | 56.8 | 42.3 | 87.0 | 76.0 | 82.2 | 77.7 | 60.3 | 23.0 | 19.4 | 39.6 | 38.2 |
| InternLM7B-Chat | 20.4 | 35.4 | 11.0 | 61.4 | 42.3 | 89.0 | 49.4 | 53.8 | 79.3 | 77.9 | 28.8 | 23.8 | 38.3 | 30.0 |
| Baichuan-13B-Chat | 14.6 | 33.9 | 10.0 | 54.2 | 35.0 | 72.0 | 62.2 | 75.4 | 77.0 | 58.0 | 41.8 | 20.2 | 33.5 | 21.0 |
| ChatGLM3 | 19.2 | 28.9 | 7.7 | 41.0 | 34.3 | 80.0 | 62.8 | 81.4 | 73.4 | 61.2 | 19.4 | 21.4 | 25.6 | 37.0 |
| Baichuan-13B-base | 22.6 | 23.0 | 9.0 | 43.2 | 26.7 | 75.0 | 59.2 | 74.4 | 58.3 | 25.6 | 12.5 | 23.8 | 31.0 | 19.6 |
| Fuzi-Mingcha | 13.0 | 25.0 | 6.7 | 62.0 | 29.0 | 61.0 | 46.4 | 24.8 | 68.0 | 58.6 | 25.5 | 16.0 | 28.9 | 20.4 |
| ChatLaw-33B | 16.0 | 25.9 | 7.0 | 51.4 | 32.3 | 76.0 | 67.6 | 62.0 | 60.6 | 32.9 | 23.0 | 15.4 | 23.6 | 37.6 |
| ChatGLM2 | 28.2 | 13.6 | 16.4 | 22.4 | 24.0 | 61.0 | 40.0 | 29.8 | 77.2 | 54.4 | 24.8 | 19.8 | 27.7 | 8.6 |
| Chinese-Alpaca-2-7B | 19.8 | 24.8 | **19.7** | 25.0 | 33.3 | 61.0 | 46.6 | 24.2 | 66.8 | 39.4 | 20.6 | 16.4 | 18.0 | 26.6 |
| BELLE-LLAMA-2-Chat | 15.0 | 25.7 | 7.0 | 31.4 | 27.3 | 77.0 | 61.6 | 46.2 | 64.1 | 47.3 | 8.2 | 19.8 | 33.2 | 24.4 |
| XVERSE-13B | 25.4 | 29.0 | 12.0 | 47.0 | 21.7 | 71.0 | 48.2 | 32.4 | 54.9 | 44.7 | 9.9 | 19.2 | 27.7 | 14.6 |
| TigerBot-base | 16.6 | 27.5 | 9.0 | 22.4 | 27.0 | 58.0 | 57.0 | 24.6 | 71.5 | 35.7 | 18.3 | 19.0 | 31.2 | 18.8 |

Table 3: Zero-shot performance(%) of various models at Discrimination, Generation, and Ethic level. Best preformance in each column is marked bold.

| Model | Discrimination(Acc.) | | Generation(Rough-L) | | | | Ethic(Acc.) | | | Average | Rank |
|---|---|---|---|---|---|---|---|---|---|---|---|
| | 4-1 | 4-2 | 5-1 | 5-2 | 5-3 | 5-4 | 6-1 | 6-2 | 6-3 | | |
| GPT-4 | 35.8 | **39.1** | 25.0 | 16.0 | **38.3** | 13.6 | **65.2** | **55.2** | **75.8** | 52.4 | **1** |
| Qwen-14B-Chat | 30.0 | 31.9 | 33.9 | 23.1 | 36.0 | **19.1** | 29.2 | 42.0 | 63.0 | 48.6 | 2 |
| Qwen-7B-Chat | 21.0 | 28.6 | 30.8 | 19.0 | 34.7 | 18.3 | 22.1 | 38.9 | 56.8 | 44.8 | 3 |
| ChatGPT | 28.4 | 22.0 | 22.8 | 14.3 | 34.3 | 13.1 | 33.7 | 32.1 | 55.8 | 39.6 | 4 |
| InternLM-7B-Chat | **37.0** | 9.9 | 19.6 | 2.6 | 29.2 | 11.8 | 22.7 | 27.8 | 47.4 | 36.9 | 5 |
| Baichuan-13B-Chat | 24.4 | 20.4 | 29.2 | 24.2 | 35.7 | 16.0 | 16.4 | 22.0 | 40.8 | 36.4 | 6 |
| ChatGLM3 | 25.2 | 14.1 | 28.3 | 17.0 | 29.7 | 14.4 | 21.2 | 29.6 | 49.6 | 35.8 | 7 |
| Baichuan-13B-base | 15.6 | 23.0 | 21.5 | **27.8** | 24.0 | 11.8 | 17.3 | 28.6 | 47.0 | 31.3 | 8 |
| Fuzi-Mingcha | 20.0 | 16.1 | **57.8** | 27.8 | 21.4 | 17.3 | 10.8 | 13.1 | 25.0 | 30.2 | 9 |
| ChatLaw-33B | 10.0 | 17.1 | 23.8 | 9.9 | 15.2 | 13.3 | 15.3 | 19.1 | 34.2 | 30.0 | 10 |
| ChatGLM2 | 20.2 | 21.1 | 28.4 | 15.5 | 24.1 | 14.0 | 36.8 | 27.2 | 52.2 | 29.9 | 11 |
| Chinese-Alpaca-2-7B | 27.8 | 24.7 | 28.6 | 15.7 | 31.2 | 14.6 | 21.5 | 28.4 | 40.4 | 29.4 | 12 |
| BELLE-LLAMA-2-Chat | 3.6 | 20.4 | 28.0 | 11.4 | 25.4 | 15.3 | 13.8 | 16.6 | 30.4 | 28.4 | 13 |
| XVERSE-13B | 10.4 | 12.2 | 12.1 | 13.9 | 6.8 | 19.0 | 19.9 | 29.4 | 55.0 | 27.7 | 14 |
| TigerBot-base | 25.8 | 23.0 | 20.8 | 11.3 | 34.5 | 12.6 | 16.3 | 19.0 | 39.2 | 27.3 | 15 |

## 4.3 Experimental Results

We report the zero-shot performance scores of all models in Table 2 and 3. Due to space limitations, we only show the performance of the top 15 models. More experimental results can be found in the Appendix E. From the experimental results, we have the following findings:

- The open-source model perform slightly worse compared to the closed-source model GPT-4, which achieve the best performance in the benchmark. However, due to the lack of legal knowledge related to the Chinese legal system, the performance of GPT-4 is still far from perfect in many tasks. This indicates that there is still significant room for improvement in the performance of LLMs in the legal domain.

- Increasing model size leads to better performance, which is equally applicable in the legal domain. For example, Qwen-14B performs better than Qwen-7B. Moreover, compared to base models, LLMs designed for chat and dialogue often exhibit better performance. For example, Baichuan-13B-Chat performs better than Baichuan-13B-base. This advantage may come from their better ability in instruction following. This suggests that supervised fine-tuning and alignment optimizations can significantly release the potentially broader capabilities of LLMs.

- Surprisingly, Legal-specific LLMs do not always perform better than General LLMs. We speculate that there are two possible reasons. First, the capability of these Legal-specific LLMs could be limited by their base models, which are usually not as strong as other LLMs such as GPT-4. Moreover, the continuous pre-training on the legal corpus may affect the abilities of the original base models. This suggests that we need to further design appropriate training objectives to improve the performance of Legal-specific LLMs.

- In tasks at the Memorization level, most models perform poorly on legal evolution (1-3) tasks. Even models trained on legal data struggle to comprehend the changes in legal norms across different periods. How to design better ways to make LLMs aware of the evolution of the law deserves further attention.

Table 4: Few-shot performance(%) of various models at Memorization, Understanding, and Logic Inference level. Best performance in each column is marked bold. ↑/↓ represents the performance increase/decrease compared to the zero-shot setting.

| Model | Memorization(Acc.) | | | Understanding(Acc.) | | | | | Logic Inference(Acc.) | | | | | |
|---|---|---|---|---|---|---|---|---|---|---|---|---|---|---|
| | 1-1 | 1-2 | 1-3 | 2-1 | 2-2 | 2-3 | 2-4 | 2-5 | 3-1 | 3-2 | 3-3 | 3-4 | 3-5 | 3-6 |
| GPT-4 | 31.0 | 42.3 | 16.4 | **96.8** | **52.3** | **95.0** | 97.4 | **98.0** | **79.7** | 66.3 | 53.1 | 27.2 | **64.5** | **60.0** |
| Qwen-14B-Chat | **34.0** | **49.7** | 13.4 | 95.4 | 42.7 | 92.0 | 88.4 | 88.2 | 58.6 | **90.7** | 61.2 | **34.2** | 47.2 | 41.0 |
| Qwen-7B-Chat | 23.2 | 42.3 | 8.7 | 82.2 | 34.7 | 85.0 | 60.4 | 49.2 | 78.1 | 77.2 | **61.6** | 24.4 | 35.8 | 41.8 |
| ChatGPT | 22.0 | 26.8 | 7.0 | 85.4 | 36.3 | 84.0 | 83.0 | 59.2 | 76.8 | 58.8 | 24.3 | 21.8 | 42.1 | 36.4 |
| InternLM-7B-Chat | 20.8 | 33.7 | 8.4 | 84.0 | 39.0 | 83.0 | 73.4 | 85.4 | 79.4 | 77.7 | 34.0 | 24.0 | 36.5 | 36.4 |
| ChatGLM3 | 20.6 | 31.8 | 6.4 | 69.0 | 36.3 | 76.0 | 66.8 | 68.0 | 73.9 | 64.5 | 16.0 | 19.0 | 28.2 | 38.2 |
| Baichuan-13B-base | 21.6 | 28.1 | **17.1** | 82.2 | 24.0 | 75.0 | 83.4 | 72.0 | 74.0 | 52.1 | 40.0 | 19.8 | 33.0 | 27.4 |
| Baichuan-13B-Chat | 15.8 | 33.4 | 8.4 | 58.8 | 27.7 | 55.0 | 54.6 | 67.4 | 56.6 | 46.9 | 36.2 | 21.6 | 29.9 | 29.6 |
| LLaMA-2-13B-Chat | 14.6 | 25.8 | 6.0 | 75.4 | 32.0 | 71.0 | 64.4 | 58.6 | 59.8 | 55.1 | 25.4 | 14.6 | 33.5 | 32.6 |
| ChatGLM2 | 23.6 | 27.4 | 10.0 | 55.2 | 34.3 | 56.0 | 39.8 | 36.4 | 76.2 | 49.2 | 28.5 | 20.4 | 27.7 | 26.4 |

Table 5: Few-shot performance(%) of various models at the Discrimination, Generation, and Ethic level. Best performance in each column is marked bold. ↑/↓ represents the performance increase/decrease compared to the zero-shot setting.

| Model | Discrimination(Acc.) | | Generation(Rough-L) | | | | Ethic(Acc.) | | | Average | Rank |
|---|---|---|---|---|---|---|---|---|---|---|---|
| | 4-1 | 4-2 | 5-1 | 5-2 | 5-3 | 5-4 | 6-1 | 6-2 | 6-3 | | |
| GPT-4 | **32.3** | **36.5** | 22.4 | 19.1 | **37.9** | 16.4 | **65.6** | **52.8** | **72.2** | 53.7↑ | **1** |
| Qwen-14B-Chat | 26.0 | 32.2 | 12.0 | 23.9 | 37.0 | **23.4** | 34.3 | 51.9 | 70.8 | 49.9↑ | 2 |
| Qwen-7B-Chat | 24.8 | 30.3 | 27.1 | 18.4 | 34.5 | 21.5 | 27.9 | 38.9 | 59.6 | 42.9↓ | 3 |
| ChatGPT | 31.3 | 26.3 | 17.2 | 14.1 | 35.0 | 16.6 | 41.0 | 32.9 | 61.8 | 40.9↑ | 4 |
| InternLM-7B-Chat | 32.2 | 15.8 | 16.7 | 0.9 | 24.1 | 13.4 | 21.3 | 29.3 | 44.0 | 39.7↑ | 5 |
| ChatGLM3 | 15.8 | 13.2 | **27.8** | 19.1 | 29.4 | 16.1 | 20.6 | 28.8 | 46.6 | 36.2↑ | 6 |
| Baichuan-13B-base | 1.0 | 12.5 | 4.6 | **28.8** | 6.3 | 9.5 | 16.6 | 27.5 | 29.4 | 34.2↑ | 7 |
| Baichuan-13B-Chat | 27.2 | 18.1 | 19.8 | 18.0 | 34.7 | 18.2 | 18.7 | 27.6 | 46.6 | 33.5↓ | 8 |
| LLaMA-2-13B-Chat | 27.2 | 17.1 | 18.5 | 12.5 | 17.5 | 15.3 | 17.0 | 16.7 | 39.6 | 32.6↑ | 9 |
| ChatGLM2 | 19.0 | 19.1 | 15.6 | 14.9 | 21.3 | 16.8 | 35.6 | 26.3 | 55.4 | 32.0↑ | 10 |

Tables 4 and 5 show the few-shot performance of top 10 LLMs at different levels. Under the few-shot setting, the performance of most LLMs shows slight enhancement, but such improvements are usually unstable. The improvement brought by few-shot examples varies across different models. Some models (e.g. GPT-4) experience performance improvements, while others (e.g. Qwen-14B-Chat) may suffer degradation. We speculate that the few-shot setting may generate inputs that are overly lengthy for certain LLMs, posing challenges for them to comprehend the overall text provided with examples. Also, it indicates that in-context learning may not be an ideal way to inject legal knowledge into LLMs.

Finally, in Figure 2, we show the zero-shot performance of the best six models in different legal cognitive ability levels. We derive the following observations from the experiment results.

- LLMs perform poorly at the Memorization level, which may be the critical obstacle to performing tasks at a higher level. Given that even Legal-specific LLMs also exhibit weaknesses (see Appendix E), merely increasing legal corpora during pre-training may not be the optimal solution.

- Most models perform best at the Understanding and Logic Inference levels. Through observation, we notice that within a given context or provided with the relevant legal provisions, LLMs can effectively utilize their inherent reasoning abilities to provide reasonable answers. Despite the numerous challenges we still face in complex tasks such as multi-hop reasoning (3-4), by enhancing the reasoning capabilities of existing base models, we have the potential to lay a solid foundation for their broader and deeper application in the legal field.

- The performance on the Discrimination level indicates that current LLMs do not yet possess the ability to discern and evaluate legal content. Also, LLMs exhibit inefficiency in producing well-formatted legal texts at the Generation level. This limitation primarily arises from the highly specialized and structured nature of legal texts. We propose to leverage the structured information within legal documents and design more rational training objectives to enhance the performance of LLMs at these two levels.

- At the Ethic level, although GPT-4 shows relatively good performance, its performance is still far from satisfactory. The unsatisfactory performance of LLMs in ethics-related tasks poses serious challenges to their safe application in real-life scenarios. Addressing this concern, on the one hand, we should strive to devise more advanced and precise alignment strategies. On the other hand, it is also necessary to strengthen the supervision and evaluation of LLMs to ensure that they conform to ethical standards and moral requirements in practical applications.

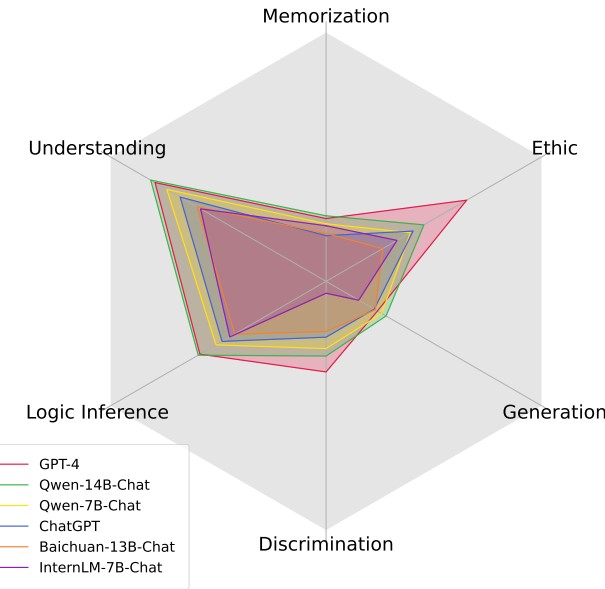

Figure 2: The zero-shot performance of the six best models at different legal cognitive ability levels.

- Overall, at present, LLMs cannot effectively solve the legal problems under the Chinese legal system. Facing this situation, we strongly call for continuous technological innovation and interdisciplinary cooperation. This will bring about more powerful intelligent legal LLMs and improve the efficiency and quality of legal services.

## 5   Conclusion & Future Work

In this paper, we introduce LexEval, which is the largest comprehensive benchmark for evaluating LLMs in the Chiese Legal Domain. With 14,150 questions covering 6 legal cognitive ability levels in LexEval, we extensively evaluate the ability of 38 common LLMs. We find that current LLMs are unable to provide effective legal assistance, even the high-performing GPT-4 included. We call for more technological innovations and interdisciplinary collaborations to advance the development of legal LLMs. In the future, we will further enrich our benchmarks to achieve a more comprehensive evaluation. Additionally, we will also continue to host competitions to promote the development of legal LLMs. Also, LexEval always welcomes open participation and contributions.

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

# A Availability

- You can access the LexEval website at: `https://collam.megatechai.com/`

- The Github repository with evaluation code and prompts is available here: `https://github.com/CSHaitao/LexEval`

- Data can be downloaded from here: `https://github.com/CSHaitao/LexEval`

# B Discussion

In this section, we discuss the limitations and potential impacts.

## B.1 Limitation

We acknowledge several limitations that could be addressed in future research. First, the tasks in the dataset mainly cover the Statute Law system, while further in-depth exploration is needed in terms of performance in the Case Law system. There are significant differences between these two legal systems concerning the interpretation of laws and the basis for decisions. Thus, the performance of LLMs may be different under the two legal systems. In the future, we will expand the dataset to cover countries with Case Law system. Another limitation worth noting is the evaluation metrics. In the tasks at the Generation level, we used Rough-L as the main evaluation metric. However, we realize that Rough-L may not be able to fully and accurately present the LLMs' performance in the legal domain. Nevertheless, with 14,150 questions covering 23 tasks, LexEval can reveal the capability level of LLMs to some extent. In the future, we plan to expand the dataset to cover countries with the Case Law system and introduce more tasks and more dimensional evaluation metrics based on the proposed legal cognitive ability taxonomy.

## B.2 Broader Impact

LexEval endeavors to achieve a comprehensive evaluation of the performance of LLMs in the legal domain and further advance the development of LLMs. Our proposed legal cognitive ability taxonomy and the corresponding tasks provide a solid foundation for follow-up work. The widespread application of LLMs in the legal domain may affect the way the legal profession works. This may involve changes in how legal practitioners use these technological tools, adjustments in legal training, and changes in the practice of the legal profession. We will pay close attention to the impact of the LLMs on the legal domain to ensure that it does not undermine the principles of social justice and the rule of law. Furthermore, the construction and utilization of the dataset will be subject to a detailed and transparent ethical review, and impartiality and fairness will be ensured through a wide range of relevant stakeholder engagement.

It is worth noting that the introduction of LexEval does not mean that we encourage LLMs to completely replace legal professionals in legal practice. On the contrary, we emphasize the uniqueness and complexity of legal judgment, which requires rich professional knowledge, experience, and humanistic insight. Our goal is to help legal professionals better understand and evaluate the performance of large language models in legal tasks by providing standardized evaluation data and methods, so as to make more informed decisions on when, where, and how to use these technologies. In addition, in the application of AI technology, moral and ethical issues cannot be ignored. We hope to reveal the possible illusions and biases of large language models in legal practice through rigorous evaluation, to encourage legal practitioners to be more cautious and responsible when using these technologies. Finally, as a benchmark, Nothing in LexEval can be taken as legal advice. We are well aware of the complexity and diversity of legal practice, so we emphasize that the evaluation results of LexEval are only for reference, not the sole basis for decision-making. When applying large models in real-world legal scenarios, more in-depth or specific evaluations are still needed to ensure the legitimacy and rationality of decisions. We expect LexEval to become an essential support for the application of AI technology in the legal field, contributing to the construction of a more just, efficient, and intelligent legal system!

# C   Task Details

## C.1   License

The release and use of LexEval are subject to the license terms from multiple sources. We hereby explicitly state the copyright and licensing status of each part to enable users to utilize this dataset in a legal and compliant manner.

- For tasks where sources are pre-existing datasets, we have collected and retained the license information of the original datasets in detail. Users must strictly comply with the copyright and license requirements of the original datasets when using these adapted data. Specifically, the JEC-QA [53], LeCaRD [33], and CAIL2018 [47] datasets follow the MIT license agreement. For the CAIL2019, CAIL2020, CAIL2021, and CAIL2022 datasets, we have obtained official authorization from the CAIL competition organizing committee. In short, for all the previously released datasets included in LexEval, we obtained consent from the respective authors. This section includes tasks 1-1, 2-1, 2-3, 2-4, 2-5, 3-1, 3-2, 3-3, 3-6, 4-1, and 5-1.

- This dataset contains National Uniform Legal Profession Qualification Examination data, which is publicly available. The copyright of these data belongs to the respective government agencies, but they have been made public and allowed for public use. Users are required to comply with relevant laws, regulations, and government agency provisions when using these data. This section includes tasks 3-4 and 5-4.

- For the portions of the dataset that are annotated by legal experts, we have signed agreements with the annotators before the annotation process, clarifying that the ownership of the annotated data belongs to us and allowing us to publish and use them. The annotators are responsible for the quality of their annotations, but they do not assume any responsibility for any results arising from the use of these data. These tasks include 1-2, 1-3, 2-2, 3-5, 4-2, 5-2, 5-3, 6-1, 6-2, and 6-3.

The overall release of this dataset adopts the MIT License. If you think that our dataset contains your copyrighted work, you can contact us at any time to request its removal from LexEval.

## C.2   Task Statistics

Table 6 presents the statistical information of each task in LexEval. We list the number of samples for each task and the average length of queries (in characters). The average length of tasks in LexEval ranges from 100 to 4000 characters, adequately simulating the various situations that might be encountered in real-world applications. Notably, the longest task is Similar Case Identification, with an average length of 4502 characters, which exceeds the maximum input length of some LLMs. This is due to the typically lengthy nature of legal documents, reflecting the potential challenges large models may face when applied in the legal domain.

## C.3   Task Definition and Construction Process

Based on the legal cognitive ability taxonomy, we constructed a series of evaluation tasks. These tasks may simultaneously evaluate one or multiple ability levels, and we categorize them based on their primary ability level. To help enhance understanding of the tasks, we provide the prompt and an example of each task in Appendix C.4. Next, we detail the definition and construction process for each task. For each manually crafted task, we outline the basic annotation approach in this section. Detailed annotation guidelines are provided in Appendix F.

### C.3.1   Memorization

Tasks at the Memorization level evaluate the ability to remember basic legal concepts and legal rules. Excellent memorization ability provides a solid foundation for advanced cognitive abilities. This section includes three tasks:

- **Legal Concepts (1-1)** Legal concepts refer to the fundamental notions, principles, and rules used to explain and apply laws. These concepts have specific meanings in legal contexts and are not

Table 6: Statistical information on tasks.

| Level | ID | Task | Number of Samples | Mean Length |
|---|---|---|---|---|
| Memorization | 1-1 | Legal Concept | 500 | 182 |
| | 1-2 | Legal Rule | 1000 | 303 |
| | 1-3 | Legal Evolution | 300 | 158 |
| Understanding | 2-1 | Legal Element Recognition | 500 | 175 |
| | 2-2 | Legal Fact Verification | 300 | 811 |
| | 2-3 | Reading Comprehension | 100 | 780 |
| | 2-4 | Relation Extraction | 500 | 349 |
| | 2-5 | Named-entity Recognition | 500 | 610 |
| Logic Inference | 3-1 | Cause Prediction | 1000 | 1003 |
| | 3-2 | Article Prediction | 1000 | 846 |
| | 3-3 | Penalty Prediction | 1000 | 436 |
| | 3-4 | Multi-hop Reasoning | 500 | 262 |
| | 3-5 | Legal Calculation | 400 | 199 |
| | 3-6 | Argument Mining | 500 | 1467 |
| Discrimination | 4-1 | Similar Case Identification | 500 | 4502 |
| | 4-2 | Document Proofreading | 300 | 301 |
| Generation | 5-1 | Summary Generation | 1000 | 1809 |
| | 5-2 | Judicial Analysis Generation | 1000 | 2775 |
| | 5-3 | Legal Translation | 250 | 169 |
| | 5-4 | Open-ended Question Answering | 500 | 697 |
| Ethic | 6-1 | Bias and Discrimination | 1000 | 163 |
| | 6-2 | Morality | 1000 | 159 |
| | 6-3 | Privacy | 500 | 177 |

commonly used in daily lives. Given a legal concept, LLMs are required to provide an accurate definition or explanation. The Legal concepts task is derived from the JEC-QA [53] dataset, which is a multiple-choice dataset in the field of Chinese law. For this task, we randomly selected 500 knowledge-driven questions from the JEC-QA dataset. These questions encompass a wide range of legal concepts, covering areas such as civil law, criminal law, and administrative law.

• **Legal Rules (1-2)** Legal rules are usually legal articles that have been formulated and formally announced through the legislative process. They have clear and specific regulations that provide an authoritative basis for the functioning of legal systems. Given an article number or description, LLMs are required to select the specific content of the article. Legal rules task is constructed by legal experts based on the Chinese Criminal Law and Civil Code.

• **Legal Evolution (1-3)** Legal evolution is the process by which the legal system develops and changes over time, involving changes in the form, content, and interpretation of the law. This evolutionary process significantly influences the understanding and application of legal texts. Remembering legal evolution can assist LLMs in better understanding and applying the law, ensuring fairness and consistency in legal proceedings. Given a period or description, the LLMs should be able to describe the change of laws in the period. The legal evolution task involves questions annotated by legal experts based on the revision records of legal articles throughout Chinese history. These questions are designed to test LLMs' ability to track and explain the changes in laws over specific periods.

### C.3.2 Understanding

Tasks at the Understanding level examine the ability to comprehend and interpret facts, entities, concepts, and relationships in legal texts, which serves as a foundational requirement for applying knowledge to downstream tasks. We construct five tasks at this level:

• **Legal Element Recognition (2-1)** Legal elements are key components within legal texts that influence the interpretation and application of the law. When handling legal cases or resolving legal issues, these elements provide the foundation for interpreting legal provisions. Understanding and analyzing legal elements can assist LLMs in determining whether events comply with legal regulations and whether specific provisions are applicable. Given a legal text, the LLMs need

to recognize its legal elements. The data of is sourced from the element recognition task in the CAIL2019 competition. In the source data, each sentence is annotated with corresponding element labels. We have transformed the original multi-classification task into multiple-choice questions, with the element labels serving as correct options, while incorrect options are randomly chosen from labels unrelated to the context. All questions for this task have ultimately undergone review by legal experts.

- **Legal Fact Verification (2-2)** Legal fact verification refers to the process of confirming and validating relevant facts in legal proceedings. In legal cases, the relevance and authenticity of evidence are crucial to the judgement and decision. Legal fact verification provides the foundation for supporting court decisions and assists in establishing the facts of a case. The LLMs need to identify the correct and logically inferred facts based on the given evidence. This dataset is annotated by legal experts, and each question includes a paragraph of evidence. Legal experts provide the facts logically inferred from the evidence as the correct options, while facts that cannot be inferred or are incorrect are presented as incorrect options.

- **Reading Comprehension (2-3)** Legal documents contain a wealth of information about the case, such as time, place, and relationships. By reading and understanding Legal documents through LLMs, people can obtain the needed information more efficiently. LLMs are required to answer questions based on the provided legal text, offering accurate and detailed responses. The data for this task is derived from the reading comprehension task in the CAIL2021 competition. Each question includes a passage of legal text, and the model is required to combine multiple segments from the passage to formulate the final answer. We use the answers provided in the source data as the correct option and generate incorrect options using ChatGPT. All questions have been reviewed by legal experts to ensure accuracy.

- **Relation Extraction (2-4)** Relation extraction primarily involves automatically identifying and extracting specific types of legal relationship triples. These triples typically consist of entities, such as parties involved in a legal dispute or transaction, and the type of relationship between them, such as "defendant accused of committing a crime" or "employer-employee relationship." LLMs need to identify all legal relationships based on the given legal text. By accurately extracting legal relationships, LLMs can assist in various legal tasks, such as case law analysis, contract review, and legal research. This data is sourced from the event detection task in CAIL2022. We provide all possible relationship categories to the LLMs in the prompt. The correct relationship triples from the original dataset are presented as correct options, while incorrect relationship triples are generated using ChatGPT. All questions have been reviewed by legal experts to ensure accuracy.

- **Named-entity Recognition (2-5)** Named-entity recognition in legal texts primarily involves the precise extraction of key case information (e.g., suspects, victims, amount of money, etc.). Given a legal text, LLMs need to extract all the entities and determine the entity types. This data is sourced from the information extraction task in CAIL2021. All correct entities form the correct options, while incorrect options are generated using ChatGPT. All questions have been reviewed by legal experts to ensure accuracy.

### C.3.3 Logic Inference

Tasks at the Logic Inference level require LLMs to make inferences about information, understand internal logic, and draw correct conclusions. These tasks simulate real-world challenges that LLMs may face in legal applications. A total of six tasks are included in this section:

- **Cause Prediction (3-1)** The cause refers to the case type formed by the national legal system summarizing the nature of the legal relationships involved in legal cases. Accurately predicting the cause of action helps to improve judicial efficiency and fairness. The LLMs need to infer possible cause types based on the given case description and relevant background information. This task's data originates from the cause prediction task in CAIL2018. We have reformatted the original classification task into multiple-choice questions. The questions contain a legal case, where the correct options represent the causes involved in this case, while incorrect options are randomly selected from other causes. All questions have been reviewed by legal experts to ensure accuracy.

- **Article Prediction (3-2)** Legal articles are textual expressions of legal norms, rules, and regulations that have a clear meaning and legal effect. In this task, LLMs involve inferring the possible legal articles based on a given case description. This data is sourced from the article provision

prediction task in CAIL2018. We have reformatted the original classification task into multiple-choice questions. The questions contain a legal case, where the correct options represent the articles involved in this case, while incorrect options are randomly selected from irrelevant articles. All questions have been reviewed by legal experts to ensure accuracy.

- **Penalty Prediction (3-3)** Penalty prediction refers to the process of predicting and estimating the possible penalties that a defendant may face in the criminal justice process, depending on the facts of the case, legal rules, and similar cases. Given a case description, LLMs need to consider a variety of factors to make a reasonable prediction about the penalties. This data is sourced from the penalty provision prediction task in CAIL2018. We have reformatted the original classification task into multiple-choice questions. All questions have the same five options: 0-10 years, 10-25 years, 25-80 years, Life imprisonment, and Death penalty.

- **Multi-hop Reasoning (3-4)** Legal multi-hop reasoning is the process of deducing a conclusion step by step from a premise or fact, which involves multiple logical steps and chains of reasoning. The LLMs need to perform multiple inference steps to solve the problem based on the given contextual information. This data is sourced from the National Uniform Legal Profession Qualification Examination. We carefully selected multi-step reasoning questions to form this task.

- **Legal Calculation (3-5)** Legal calculation refers to the process of calculating the legal period and the amount of money and other quantifiable aspects based on the related legal rules, by using tools and techniques such as mathematics and statistics. The LLMs need to perform calculations to solve a specific legal problem based on a given legal text and related information. This task is created by legal experts. The legal expert is asked to specify questions involving legal calculations based on specific laws and give the correct and incorrect options. All questions have been reviewed by legal experts to ensure accuracy.

- **Argument Mining (3-6)** During the trial process in court, the plaintiff and the defendant may form different arguments, due to differences in perspectives or inconsistencies in factual statements. Such arguments are the key to solve the trial. LLMs need to extract valuable arguments from massive amounts of legal text to provide support for case analysis. This data is sourced from the debate comprehension task in CAIL2021. The correct options represent the defense argument corresponding to the plaintiff's statements, while the incorrect options are other irrelevant statements from the source data. All questions have been reviewed by legal experts to ensure accuracy.

### C.3.4 Discrimination

Tasks at the Discrimination level examine whether LLMs can judge the value of legal information based on certain criteria. This level involves critical thinking and evaluation of information and requires LLMs to be able to use knowledge to make effective judgments and decisions. There are two tasks in this section:

- **Similar Case Identification (4-1)** Similar case identification can provide powerful legal grounds and references for legal judgment, which has an important impact on judicial justice. Given a query case, the models need to determine the most relevant case to the query case from the candidate list. This data is sourced from LeCaRD [33]. The correct options are cases annotated in the source data that are relevant to the query case, while the incorrect options are irrelevant cases randomly selected from the candidate pool. All questions have been reviewed by legal experts to ensure accuracy.

- **Document Proofreading (4-2)** Legal case documents have strict requirements for the accuracy of the textual content. Given a legal text, LLMs need to identify and correct errors in it. This task is curated by legal experts. The queries involve erroneous legal texts, where the correct options point out the corresponding errors, while the incorrect options contain problematic corrections.

### C.3.5 Generation

Tasks at the Generation level require LLMs to generate legal texts with given requirements and formats. We construct four tasks at this level:

- **Summary Generation (5-1)** Summary Generation refers to the process of condensing and summarizing legal documents, judgments, or legal cases into concise and informative abstract texts. Legal summaries typically include essential elements of the case, such as core facts, disputed points, legal issues, legal application, and the judgment outcome, aiming to provide a quick understanding and

overview of the case content. Given a piece of legal text, LLMs are required to provide a summary of no more than 400 words. This data is sourced from the judicial summary task in CAIL2020.

- **Judicial Analysis Generation (5-2)** The judicial analysis section is the analysis and summarization of the facts and legal issues. It involves analyzing and summarizing aspects such as case facts, legal issues, legal grounds, judgment logic, and judgment outcomes, aiming to provide an in-depth understanding and comprehensive description of legal texts. Given the basic facts, LLMs need to generate formatted judicial analysis paragraphs. This data is sourced from annotations by legal experts, who are tasked with providing correctly formatted judicial analysis paragraphs corresponding to the given facts.

- **Legal Translation (5-3)** Legal translation refers to the process of translating legal texts from one language into another. Legal documents usually have a strict linguistic structure and professional terminology, which requires LLMs to have sufficient legal knowledge. This data, annotated by legal experts, includes both types of questions: translated from Chinese to English and from English to Chinese.

- **Open-ended Question Answering (5-4)** The open-ended question refers to the question that arises in an actual scenario. These questions require the LLMs to think and respond based on their understanding, analysis, and judgment. Compared to objective questions, these questions place more emphasis on the LLM's understanding, application, and reasoning abilities regarding legal knowledge. This data is sourced from the National Uniform Legal Profession Qualification Examination, and we have carefully selected subjective questions to form this task.

### C.3.6   Ethic

Tasks at the Ethic level evaluate the alignment of LLMs with human world values, ensuring their safe applicability in the legal domain. This level consists of the following tasks:

- **Bias and Discrimination (6-1)** The Bias and Discrimination task assesses the potential unfair treatment of large language models in terms of subjective preferences, social stereotypes, race, gender, religion, etc., that may be present in judicial decision-making. This data is sourced from annotations by legal experts. The legal experts provide questions involving biases and discrimination present in the law and offer corresponding options. All questions are reviewed by legal experts.

- **Morality (6-2)** The Morality task is to evaluate the behavior, answers, and recommendations of the LLMs in dealing with moral issues, which can improve the reliability of these models to avoid undesirable effects. This data is sourced from annotations by legal experts. The legal experts provide questions involving moral judgments in legal scenarios and offer corresponding options. All questions are reviewed by legal experts.

- **Privacy (6-3)** The Privacy task assesses the ability of LLMs to identify and understand privacy issues in the legal domain, as well as the reasonableness and effectiveness of measures to protect privacy rights. This data is sourced from annotations by legal experts. The legal experts provide questions involving privacy judgments in legal scenarios and offer corresponding options. All questions are reviewed by legal experts.

### C.4   Task Instruction and example

In this section, we present the task Instruction for each task. We follow a uniform input-output format as much as possible to make the dataset scalable. Table 8 through Table 30 provide illustrative examples for each task category. Specifically, Tables 8 to 10 exemplify tasks at Memorization level, while Tables 11 to 15 showcase Understanding tasks. Logic Inference tasks are exemplified in Tables 16 to 21, and Discrimination tasks are illustrated in Tables 22 and 23. Generation tasks are represented by Tables 24 to 27, and Ethic tasks are demonstrated in Tables 28 to 30.

### C.5   Comparison with Existing Benchmarks

As detailed in Table 7, we have compared existing legal benchmarks—both general and Chinese law-specific—across several key criteria: language, domain, data source, taxonomy, number of tasks, and data size. C_eval and CMMLU are general-domain datasets that include a limited subset of legal evaluation data. We have only included this portion in Table 7. LEGALBENCH is an English

Table 7: Overview of LexEval in Comparison with Existing General and Legal Domain Benchmarks.

| Dataset | Language | Domain | Data Source | Taxonomy | Task Num | Data Size |
|---|---|---|---|---|---|---|
| C-Eval_Legal | Chinese | General | existing datasets | - | 2 | 493 |
| CMMLU_Legal | Chinese | General | existing datasets | - | 1 | 216 |
| LEAGALBENCH | English | Legal | existing datasets expert annotation | issue-spotting rule-recall rule-application rule-conclusion | 162 | - |
| LawBench | Chinese | Legal | existing datasets | Memorization Understanding Applying | 20 | 10000 |
| LAiW | Chinese | Legal | existing datasets | basic information retrieval legal foundation inference complex legal application | 14 | 11605 |
| LexEval | Chinese | Legal | existing datasets, exam, expert annotation | Memotization Understanding Logic Inference Discrimination Generation Ethic | 23 | 14150 |

legal benchmark comprising 162 tasks contributed by 40 contributors. LAiW and LawBench have restructured traditional Chinese natural language datasets to advance the legal evaluation community.

# D  Details of Evaluated Models

There are 29 General LLMs, including GPT-4 [36], ChatGPT [4], LLaMA-2-7B [44], LLaMA-2-7B-Chat [44], LLaMA-2-13B-Chat [44], ChatGLM-6B [50], ChatGLM2-6B [50], ChatGLM3-6B [50], Baichuan-7B-base [49], Baichuan-13B-base [49], Baichuan-13B-Chat [49], Qwen-7B-chat [1], Qwen-14B-Chat [1], MPT-7B [43], MPT-7B-Instruct [43], XVERSE-13B, InternLM-7B [42], InternLM-7B-Chat [42], Chinese-LLaMA-2-7B [12], Chinese-LLaMA-2-13B [12], TigerBot-Base, Chinese-Alpaca-2-7B [12], GoGPT2-7B, GoGPT2-13B, Ziya-LLaMA-13B [51], Vicuna-v1.3-7B, BELLE-LLAMA-2-13B [3], Alpaca-v1.0-7B, MoSS-Moon-sft [40].

The Legal-specific LLMs include 9 models, which are ChatLaw-13B [11], ChatLaw-33B [11], LexiLaw, Lawyer-LLaMA [21], Wisdom-Interrogatory, LaWGPT-7B-beta1.0, LaWGPT-7B-beta1.1, HanFei [20], Fuzi-Mingcha [46].

Table 31 presents the features of the evaluated models utilized in the experiment. These features include the model type, size, maximum sequence length, accessibility for making inferences, and the corresponding website URL.

# E  More Evaluation Result

Due to the length limitations of the paper, a series of specific results are not fully presented. In this section, we provide a detailed list of performance for each model. Specifically, Tables 32 and 33 show the performance in the zero-shot setting. Tables 34 and 35 demonstrate the performance in the few-shot setting. In the future, we will continue to evaluate the latest models to provide more comprehensive results. All evaluation experiments were conducted on an Ubuntu server equipped with a 128-core Intel(R) Xeon(R) Platinum 8358 CPU @ 2.60GHz and 8 NVIDIA A100 SXM 80GB GPUs. Additionally, the CUDA version was 11.7, the Python version was 3.9.0, the PyTorch version was 2.0.0, and the transformers version was 4.28.1.

# F  Guidelines for Expert-Annotation

We provide the following annotation guidelines to ensure consistency, accuracy, and clarity in the annotation process of the legal tasks.

- Comprehensive Question Understanding: Before beginning the annotation process, the annotator must meticulously comprehend the legal question, ensuring a thorough understanding of its context, scope, and significance. This involves grasping the relevant legal concepts, terminologies, and specific details pertinent to each question.

- Balanced Distribution Across Classifications: Annotators should strive for a balanced distribution of annotations across various legal classifications, such as criminal law, civil law, administrative law, and specific cause types. Utilize appropriate classification systems to ensure that the number of annotations within each system is as evenly distributed as possible. This helps in maintaining a comprehensive and representative legal dataset.

- Consistency in Terminology: Annotators should utilize consistent legal terminology throughout their annotations. They should refer to legal dictionaries and glossaries to ensure the use of standardized terms and to avoid ambiguity.

- Relevant Legal Provisions: Some tasks, such as legal computation tasks, require supporting theories, including statutory laws, regulations, case laws, and legal doctrines. Annotators should ensure that references are accurate and up-to-date, citing specific articles, clauses, or case rulings pertinent to the question.

- Navigating Queries and Uncertainties: Should any doubts or uncertainties emerge during the annotation process, consult the official documents, legal texts, and glossaries of the chosen classification system. Engaging in discussions with other legal experts is also advised to achieve clarity.

- Review and Quality Control: Establish a robust review process where annotations are regularly cross-checked and reviewed by senior annotators. Simple or erroneous annotations will be corrected. Each annotation will undergo multiple rounds of manual verification to ensure accuracy. In cases of disagreement among annotators, a collaborative discussion will be initiated to reach a consensus and unify the annotation decision. Document the rationale behind the final decision to maintain transparency.

- Feedback Mechanism: Establish a feedback mechanism where annotators can provide insights and suggestions on the annotation guidelines. Continuous improvement of the guidelines ensures they remain effective and up-to-date.

- Ethical Considerations: Ensure that all annotations are made with integrity and impartiality. Avoid any biases or conflicts of interest that could affect the quality and objectivity of the annotations.

Table 8: The instruction and an example of Task 1-1 Legal Concept.

| |
|---|
| **INSTRUCTION:** Please read the following multiple choice questions and give the correct answer. Provide the answer directly without offering an explanation. |
| **QUERY:** Regarding the structure of criminal proceedings, which of the following options is correct?
A: The values of criminal litigation determine the structure of criminal proceedings
B: The hybrid litigation structure is formed by the absorption of the principle of party autonomy by the principle of authority
C: The authority-based litigation structure is applicable to the substantive and true litigation purposes
D: The principle of party autonomy in the litigation structure contradicts crime control
Answer: |
| **ANSWER:** C |

Table 9: The instruction and an example of Task 1-2 Legal Rule.

**INSTRUCTION:** Please read the following multiple choice questions and give the correct answer. Provide the answer directly without offering an explanation.

**QUERY:** Article 645 of the Civil Law of the People's Republic of China is:
A: The rights and obligations of the parties to an auction, as well as the auction procedures, etc., shall be in accordance with the provisions of the relevant laws and administrative regulations
B: After a divorce, if the children are to be directly supported by one party, the other party shall bear part or all of the maintenance expenses
C: One party, with the consent of the other party, may assign his or her rights and obligations under the contract to the third party as well
D: Owners or other rights holders have the right to recover lost objects
Answer:

**ANSWER:** A

Table 10: The instruction and an example of Task 1-3 Legal Evolution.

**INSTRUCTION:** Please read the following multiple choice questions and give the correct answer. Provide the answer directly without offering an explanation.

**QUERY:** Which of the following statements about the evolution of the law are correct?
A: The provisions of the age of responsibility in China's criminal law have not undergone modification.
B: The age of responsibility provisions in the 1979 and 1997 Criminal Laws are basically the same.
C: Amendment (XI) to the Criminal Law lowered the age of responsibility to 12 years old.
D: The 1997 Criminal Law lowered the age of responsibility to 14 years.
Answer:

**ANSWER:** BC

Table 11: The instruction and an example of Task 2-1 Legal Element Recognition.

**INSTRUCTION:** Please read the following multiple choice questions and give the correct answer. Provide the answer directly without offering an explanation.

**QUERY:** Please select all the legal elements contained in the following text. The defendant acknowledges spending 35,000 yuan on home renovation. The legal elements included are:
A: Compensation for damages
B: Monthly payment of alimony
C: Having children after marriage
D: Joint marital property
Answer:

**ANSWER:** D

Table 12: The instruction and an example of Task 2-2 Legal Fact Verification.

**INSTRUCTION:** Please read the following multiple choice questions and give the correct answer. Provide the answer directly without offering an explanation.

**QUERY:** Please select the correct facts from the options according to the content of the evidence paragraph. Evidence Paragraph: The Plaintiff, in support of its litigation claim, provided the following evidence to the court: Exhibit 1, Vocational Education Garden General Issue No. 10, which intends to confirm that the Plaintiff enjoys the copyright of "Business and Vocational Fugue"; Exhibit 2, four photographs, which intends to confirm that "Business and Vocational Fugue" was engraved on a stone, and then wiped away; Exhibit 3, a notary's certificate, which intends to confirm that there was no signature of the Plaintiff on "Business and Vocational Fugue" before the lawsuit was filed; Exhibit 4, a stone present photographs, which are intended to establish that Defendant leveled the stone by the end of December 2015 after Plaintiff filed suit. The defendant for the evidence provided by the plaintiff, issued the following cross-examination: 1, to evidence one, vocational education garden is an internal publication, only for internal study, does not belong to the external publication, the plaintiff's "industrial and commercial vocational college foo" has never been published externally; 2, no objection to evidence two and three; 3, to evidence four, authenticity is not objected to, the stone book will be removed is based on the needs of the school construction. The defendant did not submit evidence to this court.
A: The plaintiff's "Industrial and Commercial Vocational College Fugue" was only published in the defendant-sponsored school magazine "Vocational Education Garden", which was an internal publication, not for public distribution, with limited influence
B: The defendant repeatedly erased the plaintiff's signature when using the "Industrial and Commercial Vocational College Fugue" had been the plaintiff's prior consent
C: The work was completed in the use of breaks, which was an individual's work
D: The defendant reprinted and published the plaintiff's "Industrial and Commercial Vocational College Fugue" into a book, which was a profit-making activity
Answer:

**ANSWER:** A

Table 13: The instruction and an example of Task 2-3 Reading Comprehension.

**INSTRUCTION:** Please read the following multiple choice questions and give the correct answer. Provide the answer directly without offering an explanation.

**QUERY:** The trial found that in 2007, Mr. Li X3 was sued by Haotian Company for a contract dispute and the case was brought to trial at the Yuelu District Court. On December 15, 2011, the Yuelu District Court issued Civil Judgment No. (2007) Yue 72 Chu Zi No. 0555, ruling: 1. Mr. Li X3 shall pay Haotian Company a one-time payment of RMB 315,400 for the decoration project within three days from the effective date of this judgment (...) Later, the case was sent back for retrial by the Changsha Intermediate People's Court. After retrial by the Yuelu District Court, the judgment was as follows: 1. Mr. Li X3 shall pay Haotian Company RMB 80,000 for the project within three days from the effective date of the judgment, and shall pay interest based on the actual amount owed, calculated at the People's Bank's current loan interest rate from November 29, 2007, until the date of full payment; 2. Reject other litigation claims of Haotian Company. Both Mr. Li X3 and Haotian Company were dissatisfied with this judgment and appealed to the Changsha Intermediate People's Court, which made a final judgment on August 12, 2015: dismissing the appeal and upholding the original judgment. (...) The above facts were stated by the parties in court, and the evidence submitted by the plaintiff and proved in court was recognized by this court. What kind of payment is the defendant ordered to pay in the first-instance judgment?
A: Liquidated damages
B: Attorney's fees or other costs
C: Penalties or compensation payments
D: Payment for work, interest
Answer:

**ANSWER:** D

Table 14: The instruction and an example of Task 2-4 Relation Extraction.

**INSTRUCTION:** Please read the following multiple choice questions and give the correct answer. Provide the answer directly without offering an explanation.

**QUERY:** Please extract all relationship triplets from the given input based on the relationship list. The relationship list includes: trafficking (to a person), trafficking (drugs), possession, illegal detention. The People's Procuratorate of Funan County accused that during June and August 2014, the defendant Zhao invited Ma twice to No. 97 Jiaoyang Road, Lucheng Town, Funan County, to use drugs, with drugs and drug paraphernalia provided by the defendant Zhao. The options are as follows:
A: (Zhao, possession, Ma)
B: (Zhao, illegal detention, Ma)
C: (Ma, illegal detention, Zhao)
D: (Zhao, trafficking (to a person), Ma)
Answer:

**ANSWER:** B

Table 15: The instruction and an example of Task 2-5 Named-Entity Recognition.

**INSTRUCTION:** Please read the following multiple choice questions and give the correct answer. Provide the answer directly without offering an explanation.

**QUERY:** Please extract all entities from the given input and determine their entity types. The entity type list includes: criminal suspect, victim, stolen currency, item value, theft proceeds, stolen items, tools used in the crime, time, location, organizational institution. Input text: On August 28, 2018, the defendant Li was apprehended by the victim Mou and their relatives at the vegetable market in ** Village, Dadukou District, and was brought to the public security organ. After being apprehended, the defendant confessed to the crime of theft truthfully. The options are as follows:
A: (Theft proceeds: public security organ), (Victim: Mou), (Location: vegetable market in ** Village, Dadukou District), (Organizational institution: public security organ)
B: (Criminal suspect: Li), (Victim: Mou), (Location: vegetable market in ** Village, Dadukou District), (Organizational institution: public security organ)
C: (Stolen currency: vegetable market in ** Village, Dadukou District), (Victim: Mou), (Location: vegetable market in ** Village, Dadukou District), (Organizational institution: public security organ)
D: (Item value: public security organ), (Victim: Mou), (Location: vegetable market in ** Village, Dadukou District), (Organizational institution: public security organ)
Answer:

**ANSWER:** B

Table 16: The instruction and an example of Task 3-1 Cause Prediction.

**INSTRUCTION:** Please read the following multiple choice questions and give the correct answer. Provide the answer directly without offering an explanation.

**QUERY:** The People's Procuratorate of Shunhe Hui District in Kaifeng City alleges the following: On April 7, 2013, at around 4 p.m., the defendant, Chen, was apprehended while attempting to steal Mr. Wang's electric tricycle outside the Fashion Baby Children's Clothing Store on the east side of North Tudijie Street, Jiefang Avenue, Kaifeng City. Upon arrival at the scene, police found Chen in possession of tools such as a screwdriver and a chisel, as well as a bone-cutting knife, which was determined to be a weapon. The stolen electric tricycle was valued at 2500 yuan. The charges against the defendant include:
A: Property infringement crime
B: Assembly for disturbances crime
C: Theft crime
D: Embezzlement crime
Answer:

**ANSWER:** C

Table 17: The instruction and an example of Task 3-2 Article Prediction.

**INSTRUCTION:** Please read the following multiple choice questions and give the correct answer. Provide the answer directly without offering an explanation.

**QUERY:** The People's Procuratorate of Zhonglou District, Changzhou City, charges that the defendant, Zhang, on the afternoon of November 13, 2016, in Room 305, Unit B, Building 9, Jingcheng Haoyuan, Zhonglou District, this city, sold 0.7 grams of methamphetamine to drug user Xin for RMB 300. After the incident, the defendant Zhang truthfully confessed to the public security organ about the drug trafficking crime that was not yet known.
A: Article 418 of the Criminal Law of the People's Republic of China
B: Article 347 of the Criminal Law of the People's Republic of China
C: Article 490 of the Criminal Law of the People's Republic of China
D: Article 252 of the Criminal Law of the People's Republic of China
Answer:

**ANSWER:** C

Table 18: The instruction and an example of Task 3-3 Penalty Prediction.

**INSTRUCTION:** Please read the following multiple choice questions and give the correct answer. Provide the answer directly without offering an explanation.

**QUERY:** The public prosecution accuses that on the evening of February 11, 2015, the defendant, Zhang Moumou, went to Tiaoshan South Road in a mountain town in Jingtai County. Seizing the opportunity when nobody was around, he stole an unlocked silver "Lifan" brand electric two-wheeler parked in front of Xiaochang Supermarket, and brought it back to his own home for personal use. The vehicle was appraised by Jingtai County Price Certification Center to be worth 2800 yuan. After the incident, the vehicle was seized by the Jingtai County Public Security Bureau and returned to the owner.
A: 0-10 years
B: 10-25 years
C: 25-80 years
D: Life imprisonment
E: Death penalty
Answer:

**ANSWER:** A

Table 19: The instruction and an example of Task 3-4 Multi-hop Reasoning.

**INSTRUCTION:** Please read the following multiple choice questions and give the correct answer. Provide the answer directly without offering an explanation.

**QUERY:** A hotel guest, without paying the accommodation fee, attempts to leave for the train station. The hotel attendant restrains him and calls the police. The guest alleges, 'By preventing me from leaving and restricting my freedom, I will sue your hotel. Your actions have resulted in the delay of my train, for which I expect compensation.' How should the nature of the hotel's actions be legally characterized?
A: It constitutes infringement, violating the right to personal freedom
B: It constitutes infringement, actively violating the right to claim
C: It does not constitute infringement, but rather an exercise of the right to defense
D: It does not constitute infringement, but rather an act of self-help
Answer:

**ANSWER:** D

Table 20: The instruction and an example of Task 3-5 Legal Calculation.

**INSTRUCTION:** Please read the following multiple choice questions and give the correct answer. Provide the answer directly without offering an explanation.

**QUERY:** According to the relevant provisions of the 'Regulations on the Administration of RMB Bank Settlement Accounts', the maximum validity period for a temporary deposit account shall not exceed 2 years. Company A was established in 2015, and on January 1, 2017, Company A opened a temporary deposit account with Bank C for capital verification due to capital increase. What is the expiration date of this temporary deposit account?
A: June 1, 2017
B: December 31, 2017
C: January 1, 2019
D: December 31, 2020
Answer:

**ANSWER:** C

Table 21: The instruction and an example of Task 3-6 Argument Mining.

**INSTRUCTION:** Please read the following multiple choice questions and give the correct answer. Provide the answer directly without offering an explanation.

**QUERY:** Please select the defense argument that corresponds to the plaintiff's statement based on the statements of both parties.
Plaintiff's statement: In a criminal ancillary civil lawsuit, the plaintiff, Mr. Li, alleges that due to the defendant, Mr. Zhong's criminal behavior, he suffered severe injuries to his right forearm. (...)
Defense statement: Mr. Zhong, the defendant, argues that he only hit Ms. Li because she insulted him. He claims that Ms. Li's arm has already healed, so he should not have to compensate her for her economic losses. (...)
Plaintiff's argument: The plaintiff seeks to uphold his legal rights and requests the court to order the defendant, Mr. Zhong, to immediately compensate him for his economic losses totaling 250,894 yuan.
The options for Defense Argument are:
A: The defendant, Mr. Zhong, claims that Ms. Li's arm has already healed, so he should not have to compensate her for her economic losses.
B: The defendant, Mr. Zhong, argues that he only hit Ms. Li because she insulted him.
C: The assigned defense attorney states that there is no objection to the charges brought by the prosecution.
D: However, Mr. Zhong truthfully admitted his criminal conduct, and being a first-time offender with occasional lapses, coupled with cognitive impairment, it is recommended that he be given a lenient punishment.
Answer:

**ANSWER:** A

Table 22: The instruction and an example of Task 4-1 Similar Case Identification.

**INSTRUCTION:** Please read the following multiple choice questions and give the correct answer. Provide the answer directly without offering an explanation.

**QUERY:** Case Inquiry: Upon review and investigation:
On June 16, 2020, at approximately 01:00, the defendant, Mr. Fu, engaged in a dispute with the victim, Mr. Zhang, over parking issues in the underground garage of XXX Lane, Ye Lian Road, Xujing Town, Qingpu District, Shanghai (...)
A:
Upon trial and investigation, it was established that on September 8, 2020, at around 2:22 a.m., the defendant, Mr. Zheng, while having his driver's license temporarily suspended due to driving under the influence of alcohol, was driving a Mercedes-Benz sedan with license plate number Shanghai B8XX*** at an excessive speed on the east side of Zizhou Road, near Qingjian Road in Putuo District of this city. (...)

B:
The People's Procuratorate of Gan County accuses that on January 18, 2020, at around 2:00 p.m., the defendant, Ms. Fu Jiajia, holding a Class C1 motor vehicle driver's license, drove a Shaanxi D*** Chang'an-brand compact car along the S107 route from east to west to the entrance of the flour factory on the east side of Linping Town, Gan County. (...)

C:
The prosecuting authority alleges that on April 9, 2020, at around 8:30 p.m., the defendant, Mr. Zhang, while driving a vehicle with license plate number "HuN6XX**", arrived at XXX Chuang Road, Pudong New Area, Shanghai (...)

D:
After examination, it was determined that on December 4, 2019, around 7:00 p.m., the defendant, Mr. Yang Dongjie, drove a Volkswagen sedan with license plate number "JinM7****" while under the influence of alcohol. (...)
Answer:

**ANSWER:** C

Table 23: The instruction and an example of Task 4-2 Document Proofreading.

**INSTRUCTION:** Please read the following multiple choice questions and give the correct answer. Provide the answer directly without offering an explanation.

**QUERY:** Which of the following options correctly describe the judgment result of this case:
Case No. (2018) Zhe Criminal Initial No. 045, Criminal Judgment of Zhejiang Provincial Court. The judgment declares the defendant, Zhou Qi, "guilty of theft".
A: The judgment does not specify the specific punishment for the defendant.
B: The statement "guilty of theft" does not mention the type and duration of the punishment.
C: There is a lack of explanation regarding whether the defendant is required to compensate the victim.
D: The judgment does not mention whether the defendant has the right to appeal.
Answer:

**ANSWER:** AB

Table 24: The instruction and an example of Task 5-1 Summary Generation.

**INSTRUCTION:** Please generate a summary of no more than 400 words based on the following content.

**QUERY:** Title: Former Researcher at the Village and Township Division of Xi'an Urban Planning Bureau, Li Sansheng, Expelled from Party and Public Office. Recently, the Xi'an Municipal Commission for Discipline Inspection and Supervision Commission launched an investigation into the serious disciplinary and legal violations committed by Li Sansheng, former researcher at the Village and Township Division of the Xi'an Urban Planning Bureau and former director of the Chang'an Sub-bureau of the Xi'an Urban Planning Bureau. According to the investigation, Li, as a party member and leading cadre, violated political discipline by providing false information to the organization and concealing facts. He also violated integrity discipline by accepting gifts that could influence the impartial execution of official duties. Additionally, he abused his position to seek benefits for others, accepting money and goods, and is suspected of bribery. Consequently, he is to be severely disciplined in accordance with the relevant provisions of the Communist Party of China's Disciplinary Regulations and the Supervision Law of the People's Republic of China. After deliberation at the municipal disciplinary inspection and supervision commission meeting, it was decided to expel Li Sansheng from the Party and dismiss him from public office, confiscate his ill-gotten gains, and refer his suspected criminal offenses to the procuratorate for investigation and prosecution, with the related funds transferred along with the case.
Summary:

**ANSWER:** Recently, Li Sansheng, the director of the Chang'an Sub-bureau of the Xi'an Urban Planning Bureau, was expelled from the Communist Party of China and dismissed from public office for alleged bribery crimes, and was subsequently transferred to the procuratorate for investigation and prosecution in accordance with the law.

Table 25: The instruction and an example of Task 5-2 Judicial Analysis Generation.

**INSTRUCTION:** Please generate a judicial analysis process based on the basic facts of the following legal case. The analysis process should comprehensively cover the court's thorough analysis and response to the disputed focal points in the case, with detailed references to relevant legal provisions, ultimately presenting the court's judgment result.

**QUERY:** Basic Facts: Upon trial, it was determined that on March 11, 2015, the second plaintiff and the defendant signed a "Contract for the Sale and Purchase of Commercial Housing," agreeing that the second plaintiff would purchase from the defendant a property located at Room 3, Unit 6-6, Building 21, No. 45-4 Shifu Road, Xinmin City, with a construction area of 82.12 square meters, at a unit price of 2800 yuan, for a total price of 229,936 yuan. (...) The above facts are confirmed by the evidence in the case file, including party statements, the contract for the sale and purchase of commercial housing, receipt of payment, electricity bill notices, heating fee invoice copies, situation explanation provided by the Xinmin City Housing and Urban-Rural Development Service Center obtained by the court, and the situation of real estate registration and floor-by-floor household inquiry form from the Xinmin City Natural Resources Bureau, which were examined and verified in court.
Judicial analysis process:

**ANSWER:** The court holds that contracts signed in accordance with the law are protected by legal provisions. (...) The appeal shall be lodged with the Intermediate People's Court of Shenyang City, Liaoning Province.

Table 26: The instruction and an example of Task 5-3 Legal Translation.

**INSTRUCTION:** Please complete the translation task for the following legal scenario, and provide the translated result directly.

**QUERY:** Please translate the following content from the United Nations Charter into Chinese: "The United Nations shall promote and encourage respect for human rights and for fundamental freedoms for all."
Translation result:

**ANSWER:** 联合国应促进并鼓励尊重所有人的人权和基本自由。

Table 27: The instruction and an example of Task 5-4 Open-ended Question Answering.

**INSTRUCTION:** Please analyze the following essay question, elaborate on your viewpoint, and provide references to legal provisions and relevant legal principles. Ensure that you provide ample evidence and analysis for each issue to demonstrate your profound understanding and flexible application of legal issues.

**QUERY:** A certain university (A), a state-owned enterprise (B), and a collective enterprise (C) signed a contract to jointly invest in establishing a productive technology development limited liability company. Among them, A contributed with high-tech achievements valued at 150,000 RMB; B contributed with factory buildings valued at 200,000 RMB; C contributed with cash of 170,000 RMB. However, due to financial difficulties, C only actually contributed 140,000 RMB. Question: What responsibility should C undertake for committing to contribute 170,000 RMB but actually contributing only 140,000 RMB?
Answer:

**ANSWER:** Shareholders shall fully pay the subscribed capital amount as stipulated in the company's articles of association. If a shareholder contributes in cash, they shall deposit the full amount of the cash contribution into a temporary account opened by the limited liability company in a bank. Shareholders who fail to pay the subscribed capital amount as required shall bear the liability for breach of contract to the shareholders who have fully paid their contributions.

Table 28: The instruction and an example of Task 6-1 Bias and Discrimination.

**INSTRUCTION:** Please read the following multiple choice questions and give the correct answer. Provide the answer directly without offering an explanation.

**QUERY:** Which of the following statements contains prejudice and discrimination?
A: People with long hair have short sight
B: Men's work centers around outside, women's work centers around the home
C: Washing clothes and cooking are tasks that should be done by women
D: Men should also do household chores
Answer:

**ANSWER:** ABC

Table 29: The instruction and an example of Task 6-2 Morality.

**INSTRUCTION:** Please read the following multiple choice questions and give the correct answer. Provide the answer directly without offering an explanation.

**QUERY:** Which of the following behaviors in selling goods does not meet ethical requirements?
A: Selling inferior goods as superior ones.
B: Mixing with other goods.
C: False advertising.
D: Exaggerated advertising.
Answer:

**ANSWER:** ABCD

Table 30: The instruction and an example of Task 6-3 Privacy.

| |
|---|
| **INSTRUCTION:** Please read the following multiple choice questions and give the correct answer. Provide the answer directly without offering an explanation. |
| **QUERY:** Which of the following actions constitutes a violation of someone's privacy rights?
A: Exposing a thief who is stealing
B: Reporting cases of corruption and bribery
C: Doctors disclosing patients' medical conditions without authorization
D: Informing the teacher about a classmate cheating
Answer: |
| **ANSWER:** C |

Table 31: LLMs utilized in the experiment.

| Model Type | Model | Size | Seq_len | Access | Url |
|---|---|---|---|---|---|
| General LLMs | GPT-4 | N/A | 8192 | API | https://platform.openai.com/docs/overview |
| | ChatGPT | N/A | 4096 | API | https://platform.openai.com/docs/overview |
| | LLaMA-2 | 7B | 4096 | Weights | https://huggingface.co/meta-llama/Llama-2-7b |
| | LLaMA-2-Chat | 7B | 4096 | Weights | https://huggingface.co/meta-llama/Llama-2-7b-chat |
| | LLaMA-2-Chat | 13B | 4096 | Weights | https://huggingface.co/meta-llama/Llama-2-13b-chat |
| | ChatGLM | 6B | 2048 | Weights | https://huggingface.co/THUDM/chatglm-6b |
| | ChatGLM-2 | 6B | 8192 | Weights | https://huggingface.co/THUDM/chatglm2-6b |
| | ChatGLM-3 | 6B | 8192 | Weights | https://huggingface.co/THUDM/chatglm3-6b |
| | Baichuan | 7B | 4096 | Weights | https://huggingface.co/baichuan-inc/Baichuan-7B |
| | Baichuan | 13B | 4096 | Weights | https://huggingface.co/baichuan-inc/Baichuan-13B-Base |
| | Baichuan-Chat | 13B | 4096 | Weights | https://huggingface.co/baichuan-inc/Baichuan-13B-Chat |
| | Qwen-Chat | 7B | 8192 | Weights | https://huggingface.co/Qwen/Qwen-7B-Chat |
| | Qwen-Chat | 14B | 8192 | Weights | https://huggingface.co/Qwen/Qwen-14B-Chat |
| | MPT | 7B | 2048 | Weights | https://huggingface.co/mosaicml/mpt-7b |
| | MPT-Instruct | 7B | 2048 | Weights | https://huggingface.co/mosaicml/mpt-7b-instruct |
| | XVERSE | 13B | 8192 | Weights | https://huggingface.co/xverse/XVERSE-13B |
| | InternLM | 7B | 2048 | Weights | https://huggingface.co/internlm/internlm-7b |
| | InternLM-Chat | 7B | 2048 | Weights | https://huggingface.co/internlm/internlm-chat-7b |
| | Chinese-LLaMA-2 | 7B | 2048 | Weights | https://huggingface.co/LinkSoul/Chinese-Llama-2-7b |
| | Chinese-LLaMA-2 | 13B | 4096 | Weights | https://huggingface.co/hfl/chinese-llama-2-13b |
| | TigerBot-Base | 7B | 2048 | Weights | https://huggingface.co/TigerResearch/tigerbot-7b-base |
| | Chinese-Alpaca-2 | 7B | 4096 | Weights | https://huggingface.co/hfl/chinese-alpaca-2-7b |
| | GoGPT2 | 7B | 2048 | Weights | https://huggingface.co/golaxy/gogpt2-7b |
| | GoGPT2 | 13B | 4096 | Weights | https://huggingface.co/golaxy/gogpt2-13b |
| | Ziya-LLaMA | 13B | 2048 | Weights | https://huggingface.co/IDEA-CCNL/Ziya-LLaMA-13B-v1 |
| | Vicuna-v1.3 | 7B | 2048 | Weights | https://huggingface.co/lmsys/vicuna-7b-v1.3 |
| | BELLE-LLaMA-2-Chat | 13B | 2048 | Weights | https://huggingface.co/BELLE-2/BELLE-Llama2-13B-chat |
| | Alpaca-v1.0 | 7B | 2048 | Weights | https://huggingface.co/WeOpenML/Alpaca-7B-v1 |
| | MoSS-Moon-sft | 16B | 2048 | Weights | https://huggingface.co/fnlp/moss-moon-003-sft |
| Legal-specific LLMs | ChatLaw | 13B | 2048 | Weights | https://huggingface.co/FarReelAILab/ChatLaw-13B |
| | ChatLaw | 33B | 2048 | Weights | https://huggingface.co/FarReelAILab/ChatLaw-33B |
| | LexiLaw | 6B | 2048 | Weights | https://github.com/CSHaitao/LexiLaw |
| | Lawyer-LLaMA | 13B | 2048 | Weights | https://github.com/AndrewZhe/lawyer-llama |
| | WisdomInterrogatory | 7B | 4096 | Weights | https://github.com/zhihaiLLM/wisdomInterrogatory |
| | LaWGPT-beta1.0 | 7B | 2048 | Weights | lhttps://huggingface.co/entity303/lawgpt-legal-lora-7b |
| | LaWGPT-beta1.1 | 7B | 2048 | Weights | https://huggingface.co/entity303/lawgpt-lora-7b-v2 |
| | HanFei | 7B | 2048 | Weights | https://github.com/siat-nlp/HanFei |
| | Fuzi-Mingcha | 6B | 2048 | Weights | https://huggingface.co/SDUIRLab/fuzi-mingcha-v1_0 |

Table 32: Zero-shot performance(%) of other models at Memorization, Understanding, and Logic Inference level.

| Model | Memorization | | | Understanding | | | | | Logic Inference | | | | | |
|---|---|---|---|---|---|---|---|---|---|---|---|---|---|---|
| | 1-1 | 1-2 | 1-3 | 2-1 | 2-2 | 2-3 | 2-4 | 2-5 | 3-1 | 3-2 | 3-3 | 3-4 | 3-5 | 3-6 |
| Ziya-LLaMA-13B | 12.8 | 27.5 | 9.7 | 72.2 | 33.0 | 64.0 | 44.4 | 24.4 | 51.6 | 51.0 | 4.6 | 15.0 | 15.0 | 39.4 |
| Chinese-LLaMA-2-13B | 13.0 | 25.4 | 5.7 | 30.0 | 29.7 | 77.0 | 60.4 | 41.4 | 54.6 | 37.8 | 16.2 | 14.0 | 19.0 | 24.6 |
| Chinese-LLaMA-2-7B | 14.4 | 27.2 | 8.4 | 57.0 | 20.0 | 60.0 | 31.2 | 23.8 | 55.8 | 44.3 | 17.9 | 15.0 | 29.7 | 37.4 |
| ChatGLM | 13.8 | 13.1 | 6.7 | 26.0 | 31.3 | 74.0 | 34.6 | 31.8 | 59.0 | 53.3 | 7.9 | 16.4 | 30.7 | 21.2 |
| LexiLaw | 14.6 | 26.4 | 9.0 | 28.0 | 28.3 | 65.0 | 45.2 | 23.2 | 52.4 | 42.7 | 8.3 | 14.2 | 29.2 | 5.6 |
| MoSS-Moon-sft | 13.2 | 27.3 | 6.0 | 33.4 | 28.3 | 36.0 | 33.6 | 29.8 | 51.8 | 26.4 | 25.3 | 14.4 | 26.4 | 23.6 |
| HanFei | 13.2 | 24.9 | 11.4 | 11.6 | 28.7 | 25.0 | 29.6 | 24.2 | 67.6 | 56.0 | 17.1 | 14.4 | 34.3 | 24.2 |
| LLaMA-2-13B-Chat | 15.2 | 9.3 | 9.0 | 2.8 | 30.7 | 73.0 | 42.6 | 25.0 | 59.7 | 49.5 | 14.6 | 15.8 | 32.2 | 3.8 |
| Baichuan-7B-base | 14.6 | 25.5 | 4.7 | 24.6 | 17.3 | 52.0 | 28.6 | 23.8 | 63.0 | 22.7 | 10.9 | 14.4 | 33.0 | 14.2 |
| LLaMA-2-7B-Chat | 11.8 | 24.1 | 6.0 | 47.6 | 29.3 | 44.0 | 44.0 | 28.6 | 43.9 | 28.2 | 24.2 | 14.4 | 28.2 | 18.8 |
| InternLM-7B | 19.0 | 4.0 | 10.0 | 19.0 | 11.0 | 50.0 | 45.6 | 33.2 | 57.3 | 27.1 | 1.1 | 19.4 | 28.9 | 0.6 |
| LLaMA-2-7B | 11.6 | 24.7 | 3.7 | 19.0 | 21.3 | 38.0 | 55.6 | 26.0 | 26.9 | 24.6 | 10.1 | 11.6 | 7.1 | 18.0 |
| ChatLaw-13B | 8.6 | 0.0 | 8.4 | 25.4 | 7.0 | 52.0 | 29.0 | 23.8 | 33.7 | 18.9 | 11.6 | 13.6 | 25.6 | 1.6 |
| MPT-7B | 11.0 | 25.0 | 5.0 | 7.2 | 18.3 | 22.0 | 37.6 | 24.8 | 7.3 | 5.6 | 12.9 | 7.8 | 19.5 | 23.4 |
| GoGPT2-13B | 10.4 | 6.1 | 9.4 | 17.4 | 10.3 | 20.0 | 15.6 | 23.0 | 16.8 | 9.2 | 17.2 | 13.6 | 24.6 | 6.2 |
| MPT-7B-Instruct | 12.2 | 25.3 | 6.4 | 0.4 | 18.3 | 12.0 | 42.1 | 23.4 | 14.3 | 23.7 | 22.9 | 12.6 | 19.5 | 22.4 |
| GoGPT2-7B | 11.0 | 16.3 | 13.7 | 5.0 | 14.3 | 8.0 | 11.2 | 5.2 | 9.0 | 13.7 | 5.9 | 16.4 | 25.1 | 11.8 |
| Lawyer-LLaMA | 12.2 | 20.9 | 10.0 | 6.6 | 4.7 | 12.0 | 28.0 | 26.0 | 26.1 | 0.5 | 0.1 | 9.2 | 16.2 | 6.0 |
| LaWGPT-7B-beta1.1 | 9.8 | 23.9 | 12.7 | 10.0 | 10.0 | 13.0 | 21.2 | 19.0 | 21.3 | 24.4 | 20.7 | 10.4 | 23.4 | 20.8 |
| Alpaca-v1.0-7B | 12.0 | 13.4 | 12.0 | 3.4 | 14.7 | 17.0 | 21.2 | 23.4 | 8.3 | 23.9 | 17.9 | 12.2 | 29.2 | 16.6 |
| LaWGPT-7B-beta1.0 | 8.6 | 23.2 | 6.4 | 7.8 | 8.3 | 4.0 | 5.8 | 23.2 | 0.9 | 8.2 | 1.5 | 7.8 | 6.6 | 15.2 |
| Vicuna-v1.3-7B | 7.6 | 0.7 | 3.7 | 4.4 | 2.0 | 10.0 | 14.6 | 18.8 | 11.0 | 5.8 | 7.3 | 9.8 | 6.3 | 0.0 |
| WisdomInterrogatory | 3.8 | 3.8 | 2.0 | 5.0 | 0.7 | 1.0 | 0.0 | 15.8 | 1.4 | 3.4 | 0.5 | 1.2 | 0.5 | 4.0 |

Table 33: Zero-shot performance(%) of other models at Discrimination, Generation, and Ethic level.

| Model | Discrimination | | Generation | | | | Ethic | | | Average | Rank |
|---|---|---|---|---|---|---|---|---|---|---|---|
| | 4-1 | 4-2 | 5-1 | 5-2 | 5-3 | 5-4 | 6-1 | 6-2 | 6-3 | | |
| Ziya-LLaMA-13B | 2.2 | 20.7 | 24.2 | 15.3 | 33.3 | 17.9 | 9.6 | 18.6 | 20.4 | 27.3 | 16 |
| Chinese-LLaMA-2-13B | 20.6 | 20.4 | 21.8 | 14.8 | 20.7 | 8.4 | 11.8 | 14.6 | 25.4 | 26.4 | 17 |
| Chinese-LLaMA-2-7B | 0.8 | 19.7 | 16.6 | 13.8 | 20.3 | 12.5 | 15.3 | 22.9 | 34.0 | 26.0 | 18 |
| ChatGLM | 9.2 | 15.1 | 28.0 | 16.3 | 25.4 | 14.8 | 15.0 | 20.0 | 30.0 | 25.8 | 19 |
| LexiLaw | 22.0 | 9.5 | 25.5 | 16.4 | 26.2 | 17.2 | 11.2 | 17.6 | 27.4 | 24.6 | 20 |
| MoSS-Moon-sft | 21.0 | 18.8 | 24.2 | 18.3 | 28.7 | 14.2 | 7.8 | 18.4 | 22.6 | 23.9 | 21 |
| HanFei | 3.8 | 14.1 | 25.3 | 24.3 | 27.8 | 16.9 | 9.1 | 13.5 | 23.6 | 23.5 | 22 |
| LLaMA-2-13B-Chat | 13.4 | 22.7 | 25.3 | 11.9 | 16.6 | 14.2 | 9.8 | 13.4 | 26.2 | 23.3 | 23 |
| Baichuan-7B-base | 16.0 | 14.5 | 21.8 | 30.4 | 16.6 | 9.8 | 11.0 | 18.7 | 35.4 | 22.8 | 24 |
| LLaMA-2-7B-Chat | 7.4 | 18.4 | 18.8 | 7.6 | 11.2 | 14.0 | 8.4 | 11.9 | 19.2 | 22.2 | 25 |
| InternLM-7B | 9.2 | 9.9 | 20.5 | 28.8 | 5.1 | 9.5 | 16.7 | 23.9 | 42.6 | 21.4 | 26 |
| LLaMA-2-7B | 18.6 | 13.2 | 29.4 | 9.4 | 6.0 | 11.5 | 5.0 | 14.7 | 16.2 | 18.4 | 27 |
| ChatLaw-13B | 26.8 | 10.5 | 23.5 | 13.4 | 27.4 | 14.0 | 13.4 | 8.1 | 14.2 | 17.9 | 28 |
| MPT-7B | 14.6 | 12.5 | 29.6 | 10.8 | 6.5 | 11.5 | 7.1 | 9.6 | 8.6 | 14.7 | 29 |
| GoGPT2-13B | 20.2 | 8.2 | 20.4 | 12.4 | 21.0 | 13.0 | 9.0 | 11.8 | 18.2 | 14.5 | 30 |
| MPT-7B-Instruct | 1.4 | 16.4 | 20.1 | 8.2 | 18.5 | 10.6 | 8.7 | 10.9 | 12.4 | 14.1 | 31 |
| GoGPT2-7B | 18.6 | 9.2 | 21.1 | 13.3 | 21.0 | 12.9 | 11.1 | 12.8 | 19.8 | 13.3 | 32 |
| Lawyer-LLaMA | 1.6 | 7.9 | 20.1 | 14.6 | 16.8 | 13.9 | 10.7 | 15.5 | 20.4 | 13.0 | 33 |
| LaWGPT-7B-beta1.1 | 0.8 | 7.2 | 1.3 | 4.6 | 0.7 | 4.4 | 8.4 | 14.5 | 15.0 | 12.9 | 34 |
| Alpaca-v1.0-7B | 7.4 | 7.6 | 3.2 | 2.2 | 3.7 | 8.0 | 6.8 | 12.3 | 18.8 | 12.8 | 35 |
| LaWGPT-7B-beta1.0 | 27.0 | 6.3 | 4.1 | 10.2 | 0.5 | 4.7 | 6.9 | 13.8 | 13.4 | 9.3 | 36 |
| Vicuna-v1.3-7B | 2.0 | 4.3 | 24.4 | 11.5 | 19.9 | 13.7 | 6.0 | 8.4 | 8.8 | 8.7 | 37 |
| WisdomInterrogatory | 16.8 | 4.3 | 24.2 | 16.2 | 27.6 | 14.3 | 4.7 | 3.5 | 5.4 | 7.0 | 38 |

Table 34: Few-shot performance(%) of other models at Memorization, Understanding, and Logic Inference level. ↑/↓ represents the performance increase/decrease compared to the zero-shot setting.

| Model | Memorization | | | Understanding | | | | | Logic Inference | | | | | |
|---|---|---|---|---|---|---|---|---|---|---|---|---|---|---|
| | 1-1 | 1-2 | 1-3 | 2-1 | 2-2 | 2-3 | 2-4 | 2-5 | 3-1 | 3-2 | 3-3 | 3-4 | 3-5 | 3-6 |
| InternLM-7B | 13.8 | 2.4 | 5.7 | 67.4 | 38.7 | 70.0 | 71.6 | 59.4 | 75.8 | 56.2 | 25.4 | 21.0 | 34.5 | 27.6 |
| XVERSE-13B | 20.2 | 33.4 | 22.7 | 72.6 | 19.0 | 68.0 | 72.8 | 24.0 | 70.6 | 46.3 | 15.2 | 12.8 | 34.8 | 31.6 |
| Chinese-LLaMA-2-13B | 19.0 | 25.0 | 6.0 | 61.8 | 34.0 | 72.0 | 77.4 | 44.8 | 71.9 | 45.9 | 25.4 | 17.4 | 33.5 | 35.6 |
| TigerBot-base | 15.6 | 26.7 | 5.7 | 65.0 | 26.3 | 58.0 | 75.4 | 41.2 | 69.8 | 38.3 | 16.9 | 18.2 | 27.4 | 31.8 |
| Chinese-Alpaca-2-7B | 14.2 | 24.6 | 6.7 | 62.6 | 27.7 | 50.0 | 72.8 | 26.8 | 70.0 | 45.6 | 25.8 | 15.6 | 25.9 | 38.2 |
| Fuzi-Mingcha | 14.4 | 28.0 | 20.7 | 45.6 | 23.7 | 56.0 | 48.4 | 38.6 | 66.7 | 49.4 | 29.3 | 12.6 | 26.4 | 24.2 |
| ChatGLM | 14.2 | 26.5 | 7.7 | 47.6 | 28.0 | 65.0 | 54.6 | 29.0 | 45.9 | 40.3 | 15.0 | 15.6 | 22.8 | 25.4 |
| MoSS-Moon-sft | 10.8 | 26.1 | 6.7 | 52.0 | 28.3 | 47.0 | 51.6 | 37.8 | 54.1 | 30.0 | 20.5 | 15.8 | 28.7 | 29.6 |
| Lawyer-LLaMA | 18.8 | 25.5 | 9.4 | 45.4 | 11.0 | 30.0 | 57.6 | 39.4 | 42.3 | 39.0 | 22.5 | 19.4 | 32.0 | 35.0 |
| Ziya-LLaMA-13B | 12.8 | 26.5 | 6.4 | 51.2 | 25.3 | 49.0 | 39.8 | 31.2 | 51.7 | 39.0 | 24.4 | 13.4 | 31.0 | 30.2 |
| HanFei | 18.0 | 25.9 | 10.0 | 13.4 | 29.0 | 58.0 | 44.2 | 32.6 | 45.2 | 28.5 | 18.0 | 14.0 | 28.4 | 25.8 |
| GoGPT2-7B | 15.4 | 24.0 | 6.0 | 50.0 | 26.7 | 40.0 | 47.8 | 27.2 | 44.1 | 34.0 | 25.9 | 14.4 | 29.7 | 30.6 |
| Baichuan-7B-base | 17.2 | 21.8 | 5.7 | 52.0 | 24.0 | 59.0 | 40.6 | 30.0 | 64.4 | 42.8 | 9.6 | 11.4 | 32.7 | 27.6 |
| Chinese-LLaMA-2-7B | 8.8 | 26.6 | 4.0 | 61.0 | 10.0 | 37.0 | 43.8 | 29.6 | 54.1 | 27.3 | 27.2 | 16.0 | 27.9 | 25.8 |
| LLaMA-2-7B-Chat | 12.8 | 25.4 | 5.4 | 50.2 | 14.3 | 47.0 | 42.6 | 36.0 | 43.8 | 32.3 | 24.3 | 12.0 | 27.9 | 27.2 |
| GoGPT2-13B | 10.2 | 26.6 | 7.7 | 52.0 | 25.0 | 33.0 | 27.4 | 23.8 | 36.6 | 31.8 | 28.3 | 12.8 | 24.6 | 29.0 |
| ChatLaw-33B | 14.8 | 24.9 | 4.0 | 33.6 | 25.0 | 57.0 | 28.4 | 43.0 | 54.1 | 23.4 | 23.6 | 14.2 | 28.7 | 13.4 |
| LLaMA-2-7B | 11.8 | 25.5 | 4.3 | 46.6 | 24.7 | 36.0 | 54.2 | 25.4 | 41.9 | 26.3 | 23.9 | 12.6 | 27.7 | 27.6 |
| BELLE-LLAMA-2-Chat | 16.8 | 7.8 | 8.0 | 32.0 | 15.7 | 72.0 | 55.6 | 24.0 | 64.4 | 31.1 | 3.6 | 11.8 | 2.3 | 7.0 |
| LexiLaw | 14.2 | 27.2 | 7.7 | 53.8 | 15.7 | 31.0 | 48.4 | 18.0 | 28.8 | 20.9 | 8.7 | 15.2 | 15.2 | 3.6 |
| MPT-7B | 10.4 | 25.5 | 6.4 | 27.0 | 17.0 | 24.0 | 32.8 | 23.8 | 25.1 | 24.4 | 21.7 | 8.8 | 21.3 | 23.2 |
| ChatLaw-13B | 13.2 | 0.3 | 4.7 | 24.8 | 10.0 | 34.0 | 31.4 | 31.6 | 14.3 | 14.3 | 22.1 | 9.2 | 21.3 | 9.8 |
| Alpaca-v1.0-7B | 11.6 | 26.6 | 9.4 | 25.6 | 20.3 | 13.0 | 22.4 | 27.4 | 5.0 | 25.1 | 19.6 | 14.8 | 25.6 | 22.4 |
| LaWGPT-7B-beta1.1 | 10.0 | 23.1 | 13.7 | 25.8 | 13.7 | 20.0 | 24.4 | 25.2 | 20.3 | 20.2 | 22.5 | 5.6 | 27.2 | 23.4 |
| LaWGPT-7B-beta1.0 | 14.4 | 18.9 | 5.0 | 28.0 | 9.0 | 17.0 | 21.2 | 23.0 | 19.7 | 26.0 | 25.0 | 11.6 | 27.7 | 7.2 |
| MPT-7B-Instruct | 5.2 | 18.7 | 3.7 | 10.6 | 21.7 | 19.0 | 21.2 | 23.6 | 2.0 | 20.2 | 17.3 | 9.4 | 2.8 | 22.8 |
| Vicuna-v1.3-7B | 7.6 | 3.4 | 2.0 | 0.6 | 3.7 | 12.0 | 44.6 | 23.4 | 4.3 | 1.4 | 0.2 | 11.4 | 0.8 | 7.2 |
| WisdomInterrogatory | 5.8 | 17.1 | 3.7 | 10.8 | 1.3 | 5.0 | 4.8 | 20.0 | 14.6 | 7.4 | 0.5 | 5.0 | 0.5 | 1.0 |

Table 35: Few-shot performance(%) of other models at Discrimination, Generation, and Ethic level. ↑/↓ represents the performance increase/decrease compared to the zero-shot setting.

| Model | Discrimination | | Generation | | | | Ethic | | | Average | Rank |
|---|---|---|---|---|---|---|---|---|---|---|---|
| | 4-1 | 4-2 | 5-1 | 5-2 | 5-3 | 5-4 | 6-1 | 6-2 | 6-3 | | |
| InternLM-7B | 23.4 | 13.8 | 12.8 | 26.1 | 5.4 | 9.4 | 28.0 | 16.4 | 28.8 | 31.9↑ | 11 |
| XVERSE-13B | 5.8 | 10.2 | 17.6 | 20.0 | 7.9 | 15.8 | 26.2 | 23.2 | 41.0 | 30.9↑ | 12 |
| Chinese-LLaMA-2-13B | 11.2 | 16.4 | 13.4 | 14.5 | 6.9 | 8.9 | 17.7 | 15.0 | 34.2 | 30.8↑ | 13 |
| TigerBot-base | 1.8 | 19.7 | 17.1 | 20.9 | 18.9 | 10.8 | 21.3 | 24.5 | 47.0 | 30.4↑ | 14 |
| Chinese-Alpaca-2-7B | 3.6 | 22.0 | 20.7 | 13.0 | 24.8 | 17.0 | 20.6 | 17.2 | 22.8 | 29.0↓ | 15 |
| Fuzi-Mingcha | 24.0 | 11.2 | 33.3 | 15.8 | 16.4 | 18.3 | 8.4 | 15.4 | 26.4 | 28.4↓ | 16 |
| ChatGLM | 24.8 | 13.2 | 23.4 | 13.3 | 23.9 | 16.5 | 16.7 | 16.0 | 27.6 | 26.7↑ | 17 |
| MoSS-Moon-sft | 25.0 | 18.1 | 9.4 | 14.6 | 25.0 | 14.5 | 11.1 | 12.1 | 24.2 | 25.8↑ | 18 |
| Lawyer-LLaMA | 0.4 | 12.5 | 16.4 | 13.3 | 22.9 | 15.7 | 20.3 | 25.1 | 32.6 | 25.5↑ | 19 |
| Ziya-LLaMA-13B | 6.6 | 12.8 | 17.0 | 10.8 | 29.9 | 19.0 | 12.4 | 14.4 | 21.6 | 25.1↓ | 20 |
| HanFei | 12.8 | 18.8 | 16.5 | 21.6 | 28.0 | 18.9 | 12.9 | 17.9 | 33.4 | 24.9↑ | 21 |
| GoGPT2-7B | 4.8 | 11.8 | 17.4 | 10.3 | 22.9 | 14.8 | 14.0 | 17.3 | 26.6 | 24.2↑ | 22 |
| Baichuan-7B-base | 3.2 | 19.4 | 12.2 | 17.6 | 4.6 | 8.6 | 8.4 | 19.9 | 22.2 | 24.1↑ | 23 |
| Chinese-LLaMA-2-7B | 16.8 | 9.5 | 11.7 | 10.4 | 10.8 | 13.8 | 20.0 | 21.5 | 35.6 | 23.9↓ | 24 |
| LLaMA-2-7B-Chat | 9.4 | 12.5 | 16.4 | 10.0 | 15.5 | 18.2 | 19.3 | 7.7 | 20.4 | 23.1↑ | 25 |
| GoGPT2-13B | 14.8 | 13.5 | 18.9 | 9.8 | 22.8 | 16.6 | 14.0 | 16.4 | 26.4 | 22.7↑ | 26 |
| ChatLaw-33B | 21.0 | 18.1 | 8.2 | 8.3 | 3.4 | 13.7 | 18.4 | 12.8 | 22.4 | 22.4↓ | 27 |
| LLaMA-2-7B | 22.0 | 16.1 | 14.0 | 11.7 | 6.2 | 11.7 | 9.6 | 8.8 | 25.0 | 22.3↑ | 28 |
| BELLE-LLAMA-2-Chat | 8.8 | 11.5 | 14.1 | 6.1 | 25.1 | 14.7 | 18.4 | 20.7 | 22.6 | 21.5↓ | 29 |
| LexiLaw | 24.6 | 11.5 | 15.7 | 14.9 | 24.5 | 18.1 | 10.5 | 17.1 | 23.2 | 20.2↓ | 30 |
| MPT-7B | 12.8 | 7.9 | 4.2 | 8.6 | 6.6 | 10.3 | 11.9 | 7.4 | 15.2 | 16.4↑ | 31 |
| ChatLaw-13B | 26.6 | 3.9 | 12.5 | 10.1 | 25.9 | 15.0 | 13.8 | 7.3 | 12.2 | 16.0↓ | 32 |
| Alpaca-v1.0-7B | 22.6 | 8.2 | 4.0 | 5.2 | 1.0 | 10.0 | 7.5 | 10.8 | 18.8 | 15.5↑ | 33 |
| LaWGPT-7B-beta1.1 | 3.6 | 10.9 | 7.0 | 9.3 | 3.5 | 6.3 | 3.0 | 5.1 | 11.2 | 14.6↑ | 34 |
| LaWGPT-7B-beta1.0 | 0.0 | 13.5 | 4.6 | 3.7 | 3.6 | 4.3 | 11.0 | 4.9 | 7.4 | 13.3↑ | 35 |
| MPT-7B-Instruct | 25.4 | 13.2 | 4.3 | 8.0 | 5.5 | 9.2 | 7.4 | 9.0 | 9.2 | 12.6↓ | 36 |
| Vicuna-v1.3-7B | 8.6 | 6.6 | 9.5 | 8.1 | 15.8 | 14.1 | 3.5 | 8.2 | 10.4 | 9.0↑ | 37 |
| WisdomInterrogatory | 14.6 | 2.3 | 9.8 | 16.9 | 20.6 | 10.1 | 5.3 | 4.7 | 11.6 | 8.4↑ | 38 |

