# OpenReview forum: "LexEval: A Comprehensive Chinese Legal Benchmark for Evaluating Large Language Models"
_NeurIPS.cc/2024/Datasets_and_Benchmarks_Track — NeurIPS 2024 Track Datasets and Benchmarks Poster_

### Official Review · Reviewer_Houv · 2024-06-27
**Comprehensive Chinese legal benchmark but the novelty should be more clearly described.**

**Rating:** 6
**Confidence:** 4
**Clarity:** Yes.

**Review:**

The paper is in overall well-written and they provide solid evaluation results.
But it maybe neccesary to present their contribution more clearly.

1. Many tasks are overlapped with LawBench paper from Fei et al, 2023 (For instance, LegalKnowledgeUnderstanding tasks). It would be necesary to describe how many tasks CoLLaM and LawBench are sharing and what are their main difference (the description from the related works section seems to be too short).

2. In continue, one of the main claim of the paper is "a standarsized comprehensive chinese legal benchmark". I generally agree yet LawBench paper also claims "a comprehensive evaluation benchmark LawBench". What is the limitation of LawBench and what makes  CoLLaM "more" comprehensive?

3. The suggested taxonomy seems interesting yet it doesn't seem to be specific to legal AI tasks. Although it is nice to have, it is unclear why it is necessary as many other works show their own taxonomy (LegalBench, LawBench, LaiW). What is the unique advantage / disadvantage of the suggested taxonomy?

Also, the pyramid in Fig.1 that placing Memorization level at the bottom seems to be misleading as the performance at this level shows very low performance (i.e. it is not necessarily easy or the basic skill required to perform other tasks).

4. It is unclear how representative the newly annotated examples are (for instance, Bias and Discrimination (6-1) Task). It would be nice if the explanation about how legal professionals selected the domains (and data) for the annotation are included in Appendix C.

5. Although it is great to include generation tasks into the benchmark, their evaluation method is too simple (ROUGE). It would be nice to include the result of human evaluation as a reference.

**Strengths:**

- CoLLaM comprises not just previous works but also newly annotated datasets.
- Seems to be more well-formatted compared to LawBench (all in Multiple-choice QA formats except some of generation tasks).
- Solid evaluation.

**Additional Feedback:**

There is no.

**Correctness:**

The "comprehesiveness", the merit of their suggested taxonomy, the task overlaps with previous works should be more clearly described.

**Documentation:**

Partially.

**Ethics:**

Yes.

**Limitations:**

Mostly.

**Opportunities For Improvement:**

Please see the review section above.

**Relation To Prior Work:**

No.

**Summary And Contributions:**

The authors propose a comprehensive Chinese legal benchmark CoLLaM that consists of 23 tasks and 13,650 questions from three sources: (1) existing datasets (CAIL< JEC-QA, LeCARD etc), (2) National Uniform Legal Profession Qualification Examination, and (3) new expert annotations.
The tasks are categoized based on suggested taxonomy including 6 types.
They have evaluated the benchmark over 38 open-sourced and commerical LLMs including GPT-4, Qwen, GLM and other LLaMA-based models.

---

> ### Author Rebuttal · Authors · 2024-08-17
>
> Dear Reviewer Houv:
>
> Thank you for your thorough and thoughtful feedback. We appreciate the opportunity to address your concerns and provide clarification on various aspects of our approach.
>
> **Comparison with Existing Legal Benchmarks**
>
> Thank you for pointing out the overlap between tasks in our CoLLaM dataset and those in the LawBench. We appreciate your suggestion to provide a more detailed comparison.
> In the revised manuscript, we will expand Section 2 to offer a more thorough comparison with both general and Chinese-specific legal benchmarks.
> As detailed in Table 1 of the attached PDF, we have compared existing legal benchmarks—both general and Chinese law-specific—across several key criteria: language, domain, data source, taxonomy, number of tasks, and data size. Given that CoLLaM is a comprehensive Chinese legal evaluation dataset, we will conduct an in-depth comparison with LaiW and LegalBench to highlight the unique contributions of each dataset.
>
> **The Unique Advantage of the Suggested Taxonomy**
>
> The LCAT is designed to address the specific cognitive abilities required for various legal tasks, offering a more targeted approach compared to existing taxonomies such as LEAGALBENCH, LawBench, and LAiW. These existing taxonomies often lack fundamental design principles. In contrast, we believe that application scenarios should be reflected through specific evaluation tasks. An effective taxonomy should capture the fundamental cognitive abilities required for legal reasoning and practice in solving legal tasks, which is the core objective of our taxonomy design. We emphasize the following key points.
>
> ***Task-Specific Insights:** The LCAT offers a detailed breakdown of cognitive abilities required for specific legal tasks. By systematically categorizing and analyzing the cognitive skills needed for different legal tasks, the LCAT provides a unified framework for organizing, expressing, and evaluating tasks in a clear and scientifically grounded manner. This approach enables a more granular understanding of how different legal skills contribute to task performance.
>
> ***Fine-Grained and Scientific Classification:** The LCAT ensures that each level represents a distinct cognitive ability, with a more balanced distribution across different levels. In contrast, existing taxonomies such as LawBench and LAiW involve broader classifications. For example, both LawBench and LAiW include a "Legal Application" level, which is quite broad. In these taxonomies, categories like Logical reasoning, Discrimination, and Generation in the LCAT might all fall under the general "Legal Application" level. By focusing on the specific cognitive abilities required to address legal tasks, the LCAT provides a more scientific and detailed classification.
>
> **Memorization level at the bottom**
>
> Thank you for pointing out this concern. We apologize for any confusion caused by the placement of the Memorization level at the bottom of Fig. 1.
> Currently, the low performance observed at the Memorization level in LLMs may be due to insufficient legal training data. We acknowledge that Memorization is not necessarily the foundational skill for other tasks. For instance, when evaluating logical reasoning or comprehension abilities, a model like GPT-4, with strong reasoning capabilities but limited Chinese legal knowledge, might better infer accurate conclusions from context. We placed the Memorization level at the bottom to reflect the idea that possessing foundational legal knowledge is a fundamental capability for a legal LLM.
> And improved memorization of legal knowledge can potentially aid in solving higher-level tasks, the reverse is not necessarily true; strong understanding or inference abilities may not inherently enhance memorization effectiveness. We understand that this placement might have been misleading. We will correct this representation in the revised version of our paper and provide a clearer explanation.
>
> **Data Representative**
>
> Thank you for pointing this out. The 6-1 task addresses issues related to bias and discrimination that may arise in legal judgments or defenses, covering various aspects such as gender, age, race, skin color, region, occupation, experience, and sexual orientation. We designed this task to be broadly representative of these critical concerns. Moreover, we will provide a detailed explanation in Appendix C about how legal professionals selected the domains and data for annotation. Additionally, we will include statistics and a comprehensive overview of the domain selection and data representation across different tasks.
>
> **Manual Evaluation of Generation tasks**
>
> Thank you for your insightful suggestions on the evaluation metrics for generative tasks. In the attached PDF, we have incorporated F1 scores for classification tasks and added BERTScore and BARTScore metrics for generative tasks. Additionally, we conducted a manual evaluation on a sampled portion of the data. We hope these enhancements offer a more comprehensive assessment of the models' performance. A detailed explanation of the manual evaluation will be provided in a separate official comment. We sincerely appreciate your valuable feedback, which has significantly improved the overall evaluation of our work.

---

> > ### Author Response · Authors · 2024-08-17
> > **In-depth Comparison with LAiW and LawBench**
> >
> > In this section, we provide a detailed comparison of LAiW, Lawbench, and CoLLaM from three aspects: data sources, taxonomy, and task setup.
> >
> > **Data Source**
> >
> > The type of data source often dictates the reliability and comprehensiveness of a dataset. LawBench and LAiW have reorganized traditional datasets to effectively utilize existing data for evaluating LLMs. However, traditional legal tasks were not specifically designed for LLMs, and their limited scope does not fully capture the capabilities of LLMs in the legal domain. Critical areas such as Legal Calculation, Legal Translation, Judicial Analysis Generation, and ethical and moral evaluation are inadequately represented in these datasets.
> >
> > In contrast, CoLLaM integrates data from existing datasets, real-world exams, and expert annotations. We not only utilize existing datasets but also collect high-quality, real-world data from the National Uniform Legal Profession Qualification Examination and engage legal experts for annotation. This diverse combination of data sources enriches CoLLaM’s tasks and provides a more comprehensive evaluation of LLMs.
> >
> > **Taxonomy**
> >
> > A robust taxonomy is essential for organizing and standardizing different tasks within a dataset. A well-structured taxonomy allows for systematic organization, clear expression, and scientific evaluation of various tasks under a unified framework.
> >
> > LawBench organizes tasks into three levels: Memory, Understanding, and Application. LAiW categorizes tasks into three types: Basic Information Retrieval, Legal Foundation Inference, and Complex Legal Application. While these taxonomies effectively structure tasks within their respective datasets, they miss critical areas in legal domain evaluation, such as ethical reasoning and generative capabilities.
> >
> > CoLLaM introduces a Legal Cognitive Ability Taxonomy (LCAT), comprising six levels: Memorization, Understanding, Logic Inference, Discrimination, Generation, and Ethics. This taxonomy is designed to reflect the comprehensive capabilities that LLMs need to develop in the legal domain, covering a broad spectrum of legal tasks. LCAT is a distinctive innovation of CoLLaM, providing a more robust and inclusive framework for task organization. This richer taxonomy lays the foundation for a thorough evaluation of LLM capabilities in the legal domain.
> >
> > **Task Setup**
> >
> > In terms of task setup and data scale, CoLLaM stands out as the largest and most comprehensive legal dataset in China, encompassing 23 tasks with a total of 13,650 datas.
> >
> > We also compared the distribution of tasks across CoLLaM, LAiW, and LawBench. CoLLaM includes 11 out of LaiW's 14 tasks, with the exceptions being Case Recognition, Case Understanding, and Legal Consultation. For LawBench, 15 out of its 20 tasks have similar counterparts in CoLLaM. Some tasks in LawBench, such as Fact-based Article Prediction and Scene-based Article Prediction, are combined into a single task in CoLLaM (Article Prediction). The tasks from LawBench not covered by CoLLaM are Dispute Focus Identification, Marital Disputes Identification, Issue Topic Identification, Trigger Word Extraction, and Consultation.
> >
> > However, CoLLaM introduces 10 unique tasks not found in the other benchmarks, including Legal Rule (1-2), Legal Evolution (1-3), Legal Fact Verification (2-2), Multi-hop Reasoning (3-4), Legal Calculation (3-5), Legal Translation (5-3), Open-ended Question Answering (5-4), Bias and Discrimination (6-1), Morality (6-2), and Privacy (6-3). These additions highlight CoLLaM’s broad scope and its capacity to evaluate a diverse range of legal capabilities.
> >
> > In addition, CoLLaM enhances standardization by converting tasks into both multiple-choice and generation formats. This approach not only standardizes evaluation methods and metrics but also provides a robust foundation for the future expansion and integration of diverse tasks. By employing these standardized formats, CoLLaM facilitates a more consistent and comprehensive assessment framework, supporting the advancement and adaptability of legal research and applications.

---

> > ### Author Response · Authors · 2024-08-17
> > **Manual Evaluation of Generation task**
> >
> > Due to cost constraints, we randomly sampled 50 queries from each task. We collected zero-shot outputs from the top six models in CoLLaM: two closed-source models (GPT-4, ChatGPT), two open-source models (Qwen-14B-Chat, Qwen-7B-Chat), and two legal-specific models (Fuzi-Mingcha, Chatlaw-33B). In total, we gathered 1,200 query-answer pairs. Each task and answer pair was annotated three times, resulting in a total of 3,600 annotations. We paid 0.22 dollars for each annotation task, resulting in a total expenditure of 792 dollars.
> >
> > Specifically, we employed three legal experts, all of whom have passed the National Uniform Legal Profession Qualification Examination. These experts are from China, including two males and one female. To ensure high-quality annotation, all annotators completed several hours of training to thoroughly understand the evaluation criteria. We assessed their performance using various examples to confirm their competency. Additionally, we provided annotators with the ground truth for the questions to help them grasp the key points of the answers more effectively. Each task and answer pair was annotated three times, and we used the Kappa statistic to measure the consistency and quality of the annotations. The Kappa values for tasks 5-1, 5-2, 5-3, and 5-4 are 0.5888, 0.754, 0.723, and 0.635, respectively, indicating the reliability of our annotations.
> >
> > We employed a 5-level pointwise scale for annotation. Annotators were provided with a (text, summary) pair or (question, answer) pair and were required to rate the quality on a 5-point scale. We adopted the Likert scale[1] as the 5-level annotation rule. For example, for the statement “The summary text adequately and briefly summarizes the core meaning of the original case” levels 1 ∼ 5 respectively represent the annotator strongly disagrees/disagrees/neutralizes/agrees/strongly agrees with the above statement. Detailed annotation standards are provided in our GitHub https://github.com/CSHaitao/CoLLaM.
> >
> > The results of the manual annotations are presented in Table 2 of the attachment PDF. We have also included the average BERTScore, BARTScore, and ROUGE-L scores for the six models on this dataset, along with their rankings for reference.
> >
> > Our findings from the experimental results are as follows:
> >
> > * GPT-4 performed the best in human evaluations, followed by Qwen-14B. The overall ranking aligns with the model performance ranking presented in Table 3 of the original paper, suggesting that CoLLaM reflects real human evaluations to some extent.
> > * For the evaluation of Generation tasks, we reported BERTScore, BARTScore, and ROUGE-L results in the table. We observed that these metrics differ to some extent from human evaluations. For instance, GPT-4, which performed best in human evaluations, ranked 4th according to ROUGE-L.
> >
> > Then, we calculated Spearman's rank correlation coefficient, Kendall rank correlation coefficient, and Pearson correlation coefficient between the three metrics and human evaluation results. The results are as follows:
> >
> > | Metrics                                  | BERTScore | BARTScore | Rouge-L |
> > | ---------------------------------------- | :-------: | :-------: | :-----: |
> > | Spearman's rank  correlation coefficient |   0.829   |    0.6    |  0.257  |
> > | Kendall rank correlation coefficient     |   0.733   |   0.467   |   0.2   |
> > | Pearson Correlation Coefficient          |   0.984   |   0.969   |  0.792  |
> >
> > The table data indicates that these metrics may not fully reflect the results of human annotations but can still provide some insight into the general capabilities and trends of the LLMs. Further research is needed to develop more effective automated evaluation methods for assessing the quality of generated text.
> >
> > Overall, BERTScore shows the highest consistency with human evaluations. Therefore, when cost constraints are a factor, we recommend using BERTScore as an alternative to human evaluations.

---

> > > ### Comment · Reviewer_Houv · 2024-08-23
> > >
> > > Thank your for your responses. They partially address my concerns, so I have adjusted the score accordingly. One clatification question: How many examples overlap between CoLLM, LAiW, and LawBench?

---

> > ### Author Response · Authors · 2024-08-25
> > **Clarification Question**
> >
> > Dear Reviewer Houv:
> >
> > Thank you for your valuable feedback. We appreciate your recognition of our efforts in addressing your concerns.
> >
> > To clarify, CoLLaM did not incorporate examples from LAiW or LawBench as its samples, so there is no direct overlap between these datasets. However, we acknowledge that in tasks where existing datasets are utilized, some overlap may occur due to the random sampling process.
> >
> > To address this, we conducted a thorough analysis of potential overlaps using two methods. The first method involved the exact matching of questions after removing instructions, punctuation, and spaces, allowing us to identify identical questions across datasets. The second method used BERTScore to calculate the similarity between questions, capturing overlaps that might not be detected through exact matching due to noise or slight variations.
> >
> > For brevity, we reported only the tasks with non-zero overlap under different metrics.
> >
> > **Table 1: Overlap between CoLLaM and LawBench.**
> > | Task  | Exact Match | BERTScore>0.85 | BERTScore>0.9 | BERTScore>0.95 |
> > | :---: | :---------: | :------------: | :-----------: | :------------: |
> > |  1_1  |      2      |       14       |      10       |       7        |
> > |  2_5  |      0      |       1        |       0       |       0        |
> > |  3_2  |      0      |       3        |       2       |       2        |
> > | Total |      2      |       18       |      12       |       9        |
> >
> > As shown in Table 1, there are 2 exact match examples and 9 examples with BERTScore similarity greater than 0.95 between CoLLaM and LawBench.
> >
> > **Table 2: Overlap between CoLLaM and LAiW.**
> > | Task  | Exact Match | BERTScore>0.85 | BERTScore>0.9 | BERTScore>0.95 |
> > | :---: | :---------: | :------------: | :-----------: | :------------: |
> > |  1_1  |      1      |       8        |       8       |       4        |
> > |  2_3  |      0      |       3        |       2       |       2        |
> > |  3_1  |      0      |       5        |       0       |       0        |
> > |  3_3  |      0      |       4        |       0       |       0        |
> > |  4_1  |      0      |       32       |      32       |       32       |
> > | Total |      1      |       52       |      42       |       38       |
> >
> > In Table 2, there is 1 exact match example and 38 examples with BERTScore similarity greater than 0.95 between CoLLaM and LAiW.
> >
> > Based on these results, it is evident that the number of overlapping examples between CoLLaM and existing datasets is extremely low, with a maximum overlap of less than 0.4%. The minimal redundancy between these datasets suggests that CoLLaM contributes fresh and valuable data to the evaluation of LLMs in the legal domain.
> >
> > Thank you for your time and effort. We hope these responses address your concerns. If you have any further questions, we would be more than happy to clarify. If our previous replies have not fully resolved your concerns, we would be glad to provide further explanations and conduct additional experiments.
> >
> > Once again, we sincerely appreciate your dedication.

---

> > > ### Comment · Reviewer_Houv · 2024-08-25
> > >
> > > Thanks for the clarification!

---

> ### Author Response · Authors · 2024-08-21
> **Sincere appreciation**
>
> Dear Reviewer Houv:
>
> Thank you very much for your time and effort in reviewing our paper. We truly appreciate your valuable feedback. If you have any further questions or concerns, please feel free to reach out. If there are no additional queries, we kindly ask you to reconsider whether our paper might be deserving of a higher score.

---

### Official Review · Reviewer_AqtW · 2024-07-11
**Review comments of submission 854**

**Rating:** 6
**Confidence:** 4
**Correctness:** Yes.
**Clarity:** Yes.

**Review:**

Strengths:

1. The CoLLaM dataset is notably extensive, comprising 13,650 questions across 23 tasks. This scale ensures a broad and thorough evaluation of LLMs in various aspects of legal cognition. The dataset integrates various data sources, including existing datasets, exam datasets, and newly annotated data by legal experts. This diversity enhances the reliability and relevance of the benchmark. The authors also have made the dataset publicly available.

2. The introduction of the Legal Cognitive Ability Taxonomy (LCAT) is a significant contribution. It provides a structured approach to categorizing and evaluating different legal cognitive abilities, which can be crucial for understanding and improving LLMs in the legal domain.

3. The manuscript discusses ethical issues involved in applying LLMs to the legal domain, highlighting potential risks such as bias and ensuring that the dataset and evaluations are ethically sound.

Weaknesses:

1. The dataset primarily focuses on the Chinese legal system, with limited evaluation of performance under other countries or other legal systems. This could restrict the benchmark's applicability.

2. The reliance on Rouge-L (Accuracy) as the main evaluation metric for generation (classification) tasks might not fully capture the quality and appropriateness of models' outputs. More nuanced metrics could provide a better assessment.

3. Some details regarding data annotation are not clearly explained, such as how the balance of legal documents containing different causes is ensured (line 186) and how sensitive information is removed from case data (line 203).

**Strengths:**

Please refer to the Review.

**Additional Feedback:**

N/A.

**Documentation:**

Yes.

**Ethics:**

No.

**Limitations:**

Yes.

**Opportunities For Improvement:**

Please refer to the weakness part of Review.

**Relation To Prior Work:**

Yes.

**Summary And Contributions:**

The manuscript  presents a new benchmark dataset designed to evaluate large language models (LLMs) specifically within the context of Chinese legal applications. The dataset includes 13,650 questions across 23 tasks, organized under a newly proposed Legal Cognitive Ability Taxonomy (LCAT) that categorizes tasks into six levels: Memorization, Understanding, Logic Inference, Discrimination, Generation, and Ethics. The authors have evaluated 38 LLMs using this benchmark, providing insights into their capabilities and limitations in the legal domain. The dataset and accompanying leaderboard are made publicly available to encourage further research and improvement in this area.

---

> ### Author Rebuttal · Authors · 2024-08-17
>
> Dear Reviewer AqtW:
>
> Thank you for your thorough and thoughtful feedback. We appreciate the opportunity to address your concerns and provide clarification on various aspects of our approach.
>
> **Focus on the Chinese legal system**
>
> Thank you for highlighting the limitation of our dataset's focus on the Chinese legal system. We acknowledge that this focus may restrict the benchmark's applicability to other legal systems. Nevertheless, the proposed LCAT is not limited to the Chinese legal system. We believe it can be extended to encompass legal tasks from other countries, as the ability levels outlined in the taxonomy are universal across different legal systems. In the future, we plan to continue expanding the legal tasks and datasets based on the LCAT to make CoLLaM a more comprehensive and versatile benchmark.
>
> **Evaluation Metrics**
>
> Thank you for your valuable feedback. We recognize that relying solely on Rouge-L (Accuracy) may not fully capture the quality and appropriateness of the models' outputs in generation and classification tasks. To address this, we have added BERTScore and BARTScore for the generative tasks, and we have also included F1 scores for the classification tasks. We believe these additional metrics will provide a more nuanced and comprehensive evaluation of the models' performance.
>
> The experimental results are detailed in Tables 5, 6, 7, and 8 in the attached PDF. Furthermore, to ensure a more thorough evaluation, we have supplemented our analysis with human annotation results for the generation tasks.  We sincerely appreciate your suggestion, as it has enabled us to enhance the robustness of our evaluation.
>
> **Some Details**
>
> Thank you for pointing out the need for more clarity regarding data annotation. We apologize for any confusion caused and will provide a more detailed explanation in the revised manuscript.
>
> To address the balance of legal documents containing various causes, we strive to ensure that the dataset is evenly distributed across different legal causes to avoid bias or long-tail effects. We achieve this balance by averaging sampling from different causes or legal domains during the data collection phase. For instance, in tasks such as Legal Rule and Legal Evolution, we ensure equal representation of civil law and criminal law. Similarly, in tasks like Cause Prediction, Penalty Prediction, and Similar Case Identification, we aim to distribute cases evenly across various causes rather than focusing on a few specific legal causes.
>
> Regarding the removal of sensitive information, we have implemented a thorough manual review process. All questions and answers are carefully examined to anonymize personal information and eliminate potentially offensive or harmful content. We will include these details in the revised version of the paper to better clarify our data annotation processes.
>
> We appreciate the time and effort you have invested in reviewing our paper. We hope the above responses address your questions and concerns effectively.

---

> > ### Author Response · Authors · 2024-08-17
> > **Manual Evaluation of Generation task**
> >
> > In this section, we describe the manual evaluation of four Generation tasks: Summary Generation (5-1), Judicial Analysis Generation (5-2), Legal Translation (5-3), and Open-ended Question Answering (5-4). The following details outline our procedures.
> >
> > Due to cost constraints, we randomly sampled 50 queries from each task. We collected zero-shot outputs from the top six models in CoLLaM: two closed-source models (GPT-4, ChatGPT), two open-source models (Qwen-14B-Chat, Qwen-7B-Chat), and two legal-specific models (Fuzi-Mingcha, Chatlaw-33B). In total, we gathered 1,200 query-answer pairs. Each task and answer pair was annotated three times, resulting in a total of 3,600 annotations. We paid 0.22 dollars for each annotation task, resulting in a total expenditure of 792 dollars.
> >
> > Specifically, we employed three legal experts, all of whom have passed the National Uniform Legal Profession Qualification Examination. These experts are from China, including two males and one female. To ensure high-quality annotation, all annotators completed several hours of training to thoroughly understand the evaluation criteria. We assessed their performance using various examples to confirm their competency. Additionally, we provided annotators with the ground truth for the questions to help them grasp the key points of the answers more effectively. Each task and answer pair was annotated three times, and we used the Kappa statistic to measure the consistency and quality of the annotations. The Kappa values for tasks 5-1, 5-2, 5-3, and 5-4 are 0.5888, 0.754, 0.723, and 0.635, respectively, indicating the reliability of our annotations.
> >
> > We employed a 5-level pointwise scale for annotation. Annotators were provided with a (text, summary) pair or (question, answer) pair and were required to rate the quality on a 5-point scale. We adopted the Likert scale[1] as the 5-level annotation rule. For example, for the statement “The summary text adequately and briefly summarizes the core meaning of the original case” levels 1 ∼ 5 respectively represent the annotator strongly disagrees/disagrees/neutralizes/agrees/strongly agrees with the above statement. Detailed annotation standards are provided in our GitHub https://github.com/CSHaitao/CoLLaM.
> >
> > The results of the manual annotations are presented in Table 2 of the attachment PDF. We have also included the average BERTScore, BARTScore, and ROUGE-L scores for the six models on this dataset, along with their rankings for reference.
> >
> > Our findings from the experimental results are as follows:
> >
> > * GPT-4 performed the best in human evaluations, followed by Qwen-14B. The overall ranking aligns with the model performance ranking presented in Table 3 of the original paper, suggesting that CoLLaM reflects real human evaluations to some extent.
> > * For the evaluation of Generation tasks, we reported BERTScore, BARTScore, and ROUGE-L results in the table. We observed that these metrics differ to some extent from human evaluations. For instance, GPT-4, which performed best in human evaluations, ranked 4th according to ROUGE-L.
> >
> > Then, we calculated Spearman's rank correlation coefficient, Kendall rank correlation coefficient, and Pearson correlation coefficient between the three metrics and human evaluation results. The results are as follows:
> >
> > | Metrics                                  | BERTScore | BARTScore | Rouge-L |
> > | ---------------------------------------- | :-------: | :-------: | :-----: |
> > | Spearman's rank  correlation coefficient |   0.829   |    0.6    |  0.257  |
> > | Kendall rank correlation coefficient     |   0.733   |   0.467   |   0.2   |
> > | Pearson Correlation Coefficient          |   0.984   |   0.969   |  0.792  |
> >
> > The table data indicates that these metrics may not fully reflect the results of human annotations but can still provide some insight into the general capabilities and trends of the LLMs. Further research is needed to develop more effective automated evaluation methods for assessing the quality of generated text.
> >
> > Overall, BERTScore shows the highest consistency with human evaluations. Therefore, when cost constraints are a factor, we recommend using BERTScore as an alternative to human evaluations.

---

> > > ### Author Response · Authors · 2024-08-29
> > > **Sincere appreciation**
> > >
> > > Dear Reviewer AqtW:
> > >
> > > Thank you very much for your thorough review and the valuable feedback you provided. We greatly appreciate the time and effort you have dedicated to evaluating our work.
> > >
> > > If you have any further questions or need additional clarification, please do not hesitate to reach out. We are more than happy to address any concerns you may have.
> > >
> > > Once again, we sincerely thank you for your invaluable contribution to improving our paper.

---

> > ### Author Response · Authors · 2024-08-21
> > **Sincere appreciation**
> >
> > Dear Reviewer AqtW:
> >
> > Thank you very much for your time and effort in reviewing our paper. We truly appreciate your valuable feedback. If you have any further questions or concerns, please feel free to reach out. If there are no additional queries, we kindly ask you to reconsider whether our paper might be deserving of a higher score.

---

### Official Review · Reviewer_NYnk · 2024-07-31
**a robust, well-structured benchmark for assessing LLMs in the legal domain, would benefit from deeper practical integration, enhanced data quality metrics, and advanced evaluation methods.**

**Rating:** 6
**Confidence:** 4
**Clarity:** Yes. The paper is overall well written

**Review:**

Quality, clarity, and originality:

The quality and clarity of the paper are high, with rigorous construction of the benchmark and thorough evaluation of multiple models. The methodology for data collection and processing is well-documented, ensuring reproducibility. The inclusion of various data sources enhances the robustness of the benchmark. The paper introduces a novel and comprehensive benchmark for evaluating LLMs in the Chinese legal domain, which is a contribution to the field.

Pros:

- Comprehensive Benchmark: CoLLaM is the largest Chinese legal evaluation dataset, covering a wide range of tasks and questions.

- Diverse Data Sources: The benchmark uses multiple data sources, including existing datasets, national exam datasets, and expert annotations, enhancing its robustness.


Cons

- Lack of Practical Correlation in Taxonomy: The proposed taxonomy, inspired by Bloom's taxonomy, is related to cognitive abilities but lacks a specific backend for the legal domain. While the paper claims that CoLLaM focuses on practical legal applications, the taxonomy seems disconnected from practical legal application and taxonomy rules. More illustration and discussion on how this taxonomy aligns with real-world legal tasks would be beneficial.

- Annotated Data Quality: The paper would benefit from reporting the inter-annotator agreement (IAA) score to ensure the quality of the annotated data. Providing IAA scores would offer more transparency and confidence in the dataset's reliability.

- Evaluation Metrics for Generation Tasks: The evaluation of generation tasks could be enhanced by incorporating more advanced metrics such as BERTScore, BARTScore, and others. Additionally, including human evaluation results would provide a more comprehensive assessment of the models' performance in generation tasks.

**Strengths:**

Comprehensive Benchmark: The introduction of CoLLaM, with its extensive dataset and a wide range of evaluation tasks, sets a new standard for legal benchmarks. It offers a thorough evaluation framework that covers various aspects of legal cognitive abilities, which is crucial for understanding the capabilities and limitations of LLMs in legal applications.

**Additional Feedback:**

N/A

**Correctness:**

Yes. The methodology for data collection and processing is well-documented, ensuring reproducibility.

**Documentation:**

No. More details can be added such as the maintenance plan.

**Limitations:**

Yes.

**Opportunities For Improvement:**

See the comments before

**Relation To Prior Work:**

Yes

**Summary And Contributions:**

This paper introduces a new benchmark designed to evaluate the performance of large language models (LLMs) in the Chinese legal domain. The benchmark, named CoLLaM, is notable for its focus on three main aspects: the introduction of a new taxonomy for legal cognitive abilities, its scale as the largest Chinese legal evaluation dataset, and the comprehensive use of various datasets, including those formatted from existing sources and newly annotated by legal experts. CoLLaM evaluates 38 LLMs across 23 tasks with 13,650 questions, addressing both fundamental legal knowledge and ethical issues. The findings highlight the current limitations of LLMs in effectively solving legal problems and provide insights for future improvements.

---

> ### Author Rebuttal · Authors · 2024-08-17
>
> Dear Reviewer NYnk:
>
> Thank you for your thorough and thoughtful feedback. We appreciate the opportunity to address your concerns and provide clarification on various aspects of our approach.
>
> **Practical Correlation in Taxonomy**
>
> Thank you for your insightful feedback regarding the practical correlation of the proposed taxonomy. We understand your concerns about the alignment of our taxonomy with real-world legal applications and taxonomy rules.
>
> To address this issue, we would like to further clarify the core concept behind our Legal Cognitive Ability Taxonomy (LCAT). The LCAT is designed to capture the cognitive abilities required for legal tasks, thereby reflecting the core demands in legal reasoning and practical application. It provides an overarching framework for categorizing and understanding the various cognitive skills necessary for specific legal tasks.
>
> A robust taxonomy is crucial for organizing and standardizing different tasks within a dataset. We believe that the effective cognitive taxonomy should distill relevant cognitive abilities from the commonalities of tasks and layer these abilities hierarchically to create a clear and structured network of relationships. In this context, the CoLLaM is closely related to real-world legal applications through specific legal tasks. The taxonomy embodies the legal cognitive abilities that LLMs need to address such tasks. The relationship between legal cognitive abilities, legal tasks, and practical applications is progressive: legal cognitive abilities are demonstrated through specific tasks and ultimately serve practical legal applications. Therefore, the CoLLaM is intricately connected to real-world legal applications rather than being disconnected. We hope this explanation clarifies how our taxonomy aligns with practical legal applications and enhances its relevance to real-world legal scenarios. Thank you for highlighting this important aspect.
>
> **Annotated Data Quality**
>
> Thank you for your valuable feedback regarding the quality of our annotated data. We appreciate your suggestion to include an inter-annotator agreement (IAA) score to ensure transparency and confidence in the dataset's reliability. However, in our context, legal experts were hired to create questions rather than annotate labels, making a traditional IAA score less applicable. To ensure the quality of our dataset, we have implemented several rigorous measures. Our gold annotators, who hold Ph.D. degrees in law, meticulously cross-check and review all generated questions. This thorough review process is designed to uphold a high standard of quality and reliability in our annotated data. Additionally, for generation tasks, we have supplemented our evaluation with human assessments and reported inter-annotator agreement scores to further enhance the quality of our annotations.
>
> **Evaluation Metrics for Generation Tasks**
>
> Thank you very much for your insightful suggestions regarding the evaluation metrics for generation tasks. In the attached PDF, we have added F1 scores for the classification tasks and included BERTScore and BARTScore metrics for the generation tasks. Additionally, we also sampled a portion of the data for manual evaluation. We hope these enhancements provide a more comprehensive assessment of the models' performance. The details of the manual evaluation will be thoroughly explained in a separate official comment. We sincerely appreciate your valuable feedback, which has helped us improve the overall evaluation of our work.
>
> We appreciate the time and effort you have invested in reviewing our paper. We hope the above responses address your questions and concerns effectively.

---

> > ### Author Response · Authors · 2024-08-17
> > **Manual Evaluation of Generation task**
> >
> > In this section, we describe the manual evaluation of four Generation tasks: Summary Generation (5-1), Judicial Analysis Generation (5-2), Legal Translation (5-3), and Open-ended Question Answering (5-4). The following details outline our procedures.
> >
> > Due to cost constraints, we randomly sampled 50 queries from each task. We collected zero-shot outputs from the top six models in CoLLaM: two closed-source models (GPT-4, ChatGPT), two open-source models (Qwen-14B-Chat, Qwen-7B-Chat), and two legal-specific models (Fuzi-Mingcha, Chatlaw-33B). In total, we gathered 1,200 query-answer pairs. Each task and answer pair was annotated three times, resulting in a total of 3,600 annotations. We paid 0.22 dollars for each annotation task, resulting in a total expenditure of 792 dollars.
> >
> > Specifically, we employed three legal experts, all of whom have passed the National Uniform Legal Profession Qualification Examination. These experts are from China, including two males and one female. To ensure high-quality annotation, all annotators completed several hours of training to thoroughly understand the evaluation criteria. We assessed their performance using various examples to confirm their competency. Additionally, we provided annotators with the ground truth for the questions to help them grasp the key points of the answers more effectively. Each task and answer pair was annotated three times, and we used the Kappa statistic to measure the consistency and quality of the annotations. The Kappa values for tasks 5-1, 5-2, 5-3, and 5-4 are 0.5888, 0.754, 0.723, and 0.635, respectively, indicating the reliability of our annotations.
> >
> > We employed a 5-level pointwise scale for annotation. Annotators were provided with a (text, summary) pair or (question, answer) pair and were required to rate the quality on a 5-point scale. We adopted the Likert scale[1] as the 5-level annotation rule. For example, for the statement “The summary text adequately and briefly summarizes the core meaning of the original case” levels 1 ∼ 5 respectively represent the annotator strongly disagrees/disagrees/neutralizes/agrees/strongly agrees with the above statement. Detailed annotation standards are provided in our GitHub https://github.com/CSHaitao/CoLLaM.
> >
> > The results of the manual annotations are presented in Table 2 of the attachment PDF. We have also included the average BERTScore, BARTScore, and ROUGE-L scores for the six models on this dataset, along with their rankings for reference.
> >
> > Our findings from the experimental results are as follows:
> >
> > * GPT-4 performed the best in human evaluations, followed by Qwen-14B. The overall ranking aligns with the model performance ranking presented in Table 3 of the original paper, suggesting that CoLLaM reflects real human evaluations to some extent.
> > * For the evaluation of Generation tasks, we reported BERTScore, BARTScore, and ROUGE-L results in the table. We observed that these metrics differ to some extent from human evaluations. For instance, GPT-4, which performed best in human evaluations, ranked 4th according to ROUGE-L.
> >
> > Then, we calculated Spearman's rank correlation coefficient, Kendall rank correlation coefficient, and Pearson correlation coefficient between the three metrics and human evaluation results. The results are as follows:
> >
> > | Metrics                                  | BERTScore | BARTScore | Rouge-L |
> > | ---------------------------------------- | :-------: | :-------: | :-----: |
> > | Spearman's rank  correlation coefficient |   0.829   |    0.6    |  0.257  |
> > | Kendall rank correlation coefficient     |   0.733   |   0.467   |   0.2   |
> > | Pearson Correlation Coefficient          |   0.984   |   0.969   |  0.792  |
> >
> > The table data indicates that these metrics may not fully reflect the results of human annotations but can still provide some insight into the general capabilities and trends of the LLMs. Further research is needed to develop more effective automated evaluation methods for assessing the quality of generated text.
> >
> > Overall, BERTScore shows the highest consistency with human evaluations. Therefore, when cost constraints are a factor, we recommend using BERTScore as an alternative to human evaluations.

---

> > ### Author Response · Authors · 2024-08-21
> > **Sincere appreciation**
> >
> > Dear Reviewer NYnk:
> >
> > Thank you very much for your time and effort in reviewing our paper. We truly appreciate your valuable feedback. If you have any further questions or concerns, please feel free to reach out. If there are no additional queries, we kindly ask you to reconsider whether our paper might be deserving of a higher score.

---

> > > ### Author Response · Authors · 2024-08-29
> > > **Sincere appreciation**
> > >
> > > Dear Reviewer NYnk:
> > >
> > > Thank you very much for your thorough review and the valuable feedback you provided. We greatly appreciate the time and effort you have dedicated to evaluating our work.
> > >
> > > If you have any further questions or need additional clarification, please do not hesitate to reach out. We are more than happy to address any concerns you may have.
> > >
> > > Once again, we sincerely thank you for your invaluable contribution to improving our paper

---

### Official Review · Reviewer_tqv1 · 2024-08-01
**Needs minor revisions**

**Rating:** 7
**Confidence:** 5
**Correctness:** The design and evaluation procedures …

**Review:**

Benchmarking is critical aspect of legal data analytics, and the authors make an effort in the same direction, by creating a diverse dataset for evaluating the legal capabilities of LLMs over Chinese law.

Regarding the work:-
Pros:
1. Sound design philosophy
2. Extensive data cleaning and annotation by legal experts
3. A large number of LLMs tested in both zero and few-shot settings
4. Tasks are meaningful and needed in real-world legal scenarios

Cons:
1. Converting to multiple-choice makes many of the tasks a lot easier than the real-world application
2. Related to the above point; could have used the real-world scenario (no multiple choice) and human evaluation for some of the tasks, to bridge the gap between experimental results and real-world performance
3. No fine-tuning experiments either with LLMs or smaller models

Regarding the paper:-
Pros:
1. Paper is very well-written and orgnanized
2. Sufficient examples of the tasks have been provided in the appendix along with the prompts, etc.

Cons:
1. I think this paper needs a section/sub-section for more detailed comparison with other existing legal benchmarks

**Strengths:**

1. Sound design philosophy
2. Extensive data cleaning and annotation by legal experts
3. A large number of LLMs tested in both zero and few-shot settings
4. Tasks are meaningful and needed in real-world legal scenarios
5. Paper is very well-written and orgnanized
6. Sufficient examples of the tasks have been provided in the appendix, along with the prompts, etc.

**Additional Feedback:**

N/A

**Clarity:**

The paper is well written, concise and easy to understand. The supplementary text is informative.

**Documentation:**

Sufficient details have been provided

**Ethics:**

No ethical concerns

**Limitations:**

The authors have sufficiently addressed the limitations of their work

**Opportunities For Improvement:**

1. This may not be a criticism of this paper in particular, but the usual design philosophy also followed in prior works. In the legal domain, converting open-ended answering to a multiple-choice format can make the task a lot easier, especially for tasks like legal rule, article prediction, similar case identification, etc. This is because, practically, these tasks can have a large label space and boiling it down to 4/5 choices makes the problem a lot simpler than it actually is. While this helps in designing uniform prompts and evaluating on an even scale, I believe that the simplicity of the tasks in MCQ format is not practical for testing real-world legal knowledge.

2. Related to the above point, the authors could have, at least for a few tasks, conducted an experiment with the real-world setting (no choices). While this could lead to generations that are difficult to evaluate using standard metrics, the experts could have been used to validate the answers (should be much less costly compared to annotation).

3. I also believe that while this is a benchmark for generation-based LLMs, it is still essential to compare the performance with discriminative BERT-based models, especially because many prior works in the legal domain have shown that LLMs, at least without fine-tuning, are often not at par with fine-tuned BERT-based models. I believe that this would show up as a bigger gap in performance if the real-world setting described above is used (no MCQ format).

4. There is only a single paragraph under Section 2 that compares CoLLaM with prior legal benchmarks --- I believe that this needs a more dedicated comparison, given the large number of legal benchmarks released recently (both in general as well as Chinese law specific). Is CoLLaM indeed better than these benchmarks? Although the design policy of CoLLaM seems to be the biggest distinguishing factor, this needs to be brought out more to help the average reader.

**Relation To Prior Work:**

Could have been clarified a bit more:

There is only a single paragraph under Section 2 that compares CoLLaM with prior legal benchmarks --- I believe that this needs a more dedicated comparison, given the large number of legal benchmarks released recently (both in general as well as Chinese law specific). Is CoLLaM indeed better than these benchmarks? Although the design policy of CoLLaM seems to be the biggest distinguishing factor, this needs to be brought out more to help the average reader.

**Summary And Contributions:**

In this work, the authors create a legal benchmark in Chinese for evaluating legal capabilities of LLMs. The authors employ a sound design policy, by choosing 23 different tasks from 6 different levels of legal cognition -- memorization, understanding, logic inference, discrimination, generation and ethics. They construct the datasets either by modifying existing datasets to suit LLM generation, from questions in a law exam, or by creating new datasets with the help of multiple legal experts. They evaluate with multiple LLMs -- open, closed and law-specific, under both zero and few-shot settings. Their results show that while LLMs are good at tasks at the legal understanding and logic inference levels, they fall short at the discrimination, generation and memorization levels.

---

> ### Author Rebuttal · Authors · 2024-08-17
>
> Dear Reviewer tqv1:
>
> Thank you for your thorough and thoughtful feedback. We appreciate the opportunity to address your concerns and provide clarification on various aspects of our approach.
>
> **Regarding the Design of Multiple-Choice Questions and Experimenting with Generation Tasks**
>
> Thank you for your valuable feedback. We recognize that converting tasks to a multiple-choice format can reduce their difficulty. In the early stages of our dataset construction, we experimented with designing some tasks as open-ended questions. For instance, in the Legal Rule task (1-2), a more challenging approach would be to require the model to fully recite the relevant legal provisions. However, during our tests, we found that current LLMs, lacking sufficient legal training data and knowledge, performed poorly in open-ended formats. Given the current limitations of legal LLMs, overly challenging tasks may not effectively distinguish between different models or contribute to their development. Additionally, as you pointed out, using multiple-choice questions facilitates evaluation on a uniform scale. Considering these factors, we ultimately opted to convert the tasks into a multiple-choice format.
>
> While multiple-choice questions may not fully capture the complexity of real-world applications, they are still a reliable way to assess an LLM's legal capabilities, much like how standardized exams in human society reflect varying levels of proficiency. However, as the capabilities of legal LLMs improve, we plan to explore offering both multiple-choice and open-ended formats for certain tasks to better align with real-world needs.
>
> Moreover, four tasks in our dataset go beyond the multiple-choice format to simulate real-world scenarios. These include Summary Generation (5-1), Judicial Analysis Generation (5-2), Legal Translation (5-3), and Open-ended Question Answering (5-4). While our initial evaluation only uses the Rouge-L metric, we recognize that this alone does not provide a complete picture. To address this, we have expanded our assessment with additional metrics, including BERTScore, BARTScore, and human evaluation, as detailed in Table 2 of the attached PDF. A thorough explanation of the criteria and methodology used in the human evaluation will be provided in subsequent official comments. We hope these enhanced evaluation results offer a clearer and more nuanced understanding of our work.
>
> **Comparison with Fine-tuned BERT-based Models**
>
> Thank you for your insightful comment. We appreciate your suggestion to compare the performance of LLMs with fine-tuned BERT-based models, especially given that prior studies in the legal domain have highlighted the disparity between the performance of fine-tuned BERT models and LLMs. However, since the CoLLaM dataset is designed as a benchmarking dataset and not for model training, it does not provide the necessary resources to fine-tune a BERT model directly on it. To address this, we conducted an additional experiment using the JEC-QA dataset, where we fine-tuned Chinese-BERT and evaluated its performance in a multiple-choice format. The results of fine-tuning BERT are presented in Tables 3 and 4 of the attached PDF. Specific implementation details and further analysis will be provided in a subsequent official comment.
>
> **Comparison with Existing Legal Benchmarks**
>
> We agree that a more detailed comparison between CoLLaM and other recent legal benchmarks is necessary. In the revised manuscript, we will expand Section 2 to offer a more thorough comparison with both general and Chinese-specific legal benchmarks.
> As detailed in Table 1 of the attached PDF, we have compared existing legal benchmarks—both general and Chinese law-specific—across several key criteria: language, domain, data source, taxonomy, number of tasks, and data size.
> C_eval and CMMLU are general-domain datasets that include a limited subset of legal evaluation data. We have only included this portion in Table 1. LEGALBENCH is an English legal benchmark comprising 162 tasks contributed by 40 contributors. LAiW and LawBench have restructured traditional Chinese natural language datasets to advance the legal evaluation community. Given that CoLLaM is a comprehensive Chinese legal evaluation dataset, we will conduct an in-depth comparison with LAiW and LawBench in other official comments to highlight the unique contributions of each dataset.
>
> We appreciate the time and effort you have invested in reviewing our paper. We hope the above responses address your questions and concerns effectively.

---

> > ### Comment · Reviewer_tqv1 · 2024-08-19
> > **Increased the score of the paper**
> >
> > I am satisfied with the author's replies to my comments. I have increased the score.

---

> > > ### Author Response · Authors · 2024-08-20
> > > **Thanks**
> > >
> > > Thank you very much for your positive feedback and for taking the time to review our revisions. We greatly appreciate your support and the valuable comments that helped improve our paper. Thank you once again for your efforts and consideration.

---

> ### Author Response · Authors · 2024-08-17
> **Implementation details and Further Analysis of BERT-based Models**
>
> In this section, we provide a detailed implementation and analysis of fine-tuned BERT-based models.
>
> **Implementation details**
>
> We utilized the JEC-QA[1] dataset as our training set. JEC-QA is currently the largest Chinese multiple-choice dataset in the legal domain, consisting of 21,076 training samples. To prevent data leakage, we excluded any samples that overlap with the task (1-1) in CoLLaM. Additionally, we randomly selected 20% of the data to serve as a validation set.
>
> Regarding the output format, some questions in the dataset have multiple correct answers, making them more complex than simple multiple-choice questions. We follow to the setting described in the original JEC-QA paper. For all questions, the model outputs a score vector with a length of 2n-1, where n represents the number of options. These scores correspond to the probability of each possible combination of options.
>
> For our experiments, we fine-tuned the chinese-bert-wwm[2] from Huggingface. We fine-tuned the model for up to 6 epochs. The fine-tuning process employed the AdamW optimizer with a learning rate of 5e-6 and a batch size of 4. We used a linear learning rate schedule with a warmup proportion of 0.1. The best-performing checkpoint on the validation set was selected for testing.
>
> **Experimental Results**
>
> The experimental results are presented in Tables 3 and 4. From these results, we observed the following:
>
> * Fine-tuned BERT-based models performed poorly in the legal domain. Specifically, for in-domain data (task 1-1), BERT achieved an accuracy of 17.2%, which aligns with the results reported in the original JEC-QA paper. In contrast, GPT achieved an accuracy of 27.2%, significantly outperforming the fine-tuned BERT model.
> * For other out-of-domain data, the fine-tuned BERT model also underperformed compared to LLMs. The only exception was in the 4-1 task, where BERT outperformed GPT-4. This may be due to the influence of long context in this task, which may have impacted GPT's judgment.
>
> These findings do not fully align with the research by Thanmay Jayakumar et al. [3], which may be due to several factors:
>
> * Firstly, the dataset used by Thanmay Jayakumar et al. is a subset of LEDGAR from the LexGLUE benchmark. This task involves classifying a given supply contract into one of 100 EDGAR topic labels. In this context, a fine-tuned BERT-based model achieved a micro F1 score of 88.2. The task relies to some extent on semantic classification, which enables fine-tuned BERT-based models to perform well, particularly when provided with sufficient training data. As a result, it is possible for BERT-based models to outperform LLMs in this specific scenario.
> * In contrast, the tasks in the CoLLaM dataset extend beyond simple legal text classification and demand a certain level of legal knowledge and reasoning ability. This poses a significant challenge for BERT-based models, which often lack the advanced reasoning capabilities required for tasks involving complex reasoning, leading to poorer performance. Additionally, the length of legal documents can impact the performance of BERT-based models due to their limited context window. In contrast, LLMs benefit from greater context awareness, larger parameter quantities, extensive pre-trained knowledge, and superior generalization capabilities.
>
> These experimental results indicate that, unlike traditional natural language processing datasets in the legal domain, CoLLaM is a demanding benchmark specifically designed for LLMs. The poor performance of BERT-based models highlights the dataset's difficulty and underscores its effectiveness in assessing the capabilities of LLMs within the legal domain. This reinforces the notion that CoLLaM serves as a rigorous test for evaluating LLMs, particularly in areas requiring in-depth legal reasoning and contextual understanding.
>
> As the field evolves, the continued development and use of such benchmarks will be crucial for pushing the boundaries of model performance and advancing our understanding of their capabilities in complex legal contexts. We believe it adds valuable context to the ongoing discussion on the efficacy of different model architectures in legal AI tasks. Thank you again for your constructive feedback.
>
> [1] Zhong H, Xiao C, Tu C, et al. JEC-QA: a legal-domain question answering dataset[C]//Proceedings of the AAAI conference on artificial intelligence. 2020, 34(05): 9701-9708.
>
> [2]https://huggingface.co/google-bert/bert-base-chinese
>
> [3] Thanmay Jayakumar, Fauzan Farooqui, and Luqman Farooqui. 2023. Large Language Models are legal but they are not: Making the case for a powerful LegalLLM. In Proceedings of the Natural Legal Language Processing Workshop 2023, pages 223–229, Singapore. Association for Computational Linguistics

---

> ### Author Response · Authors · 2024-08-17
> **In-depth Comparison with LAiW and LawBench**
>
> In this section, we provide a detailed comparison of LAiW, Lawbench, and CoLLaM from three aspects: data sources, taxonomy, and task setup.
>
> **Data Source**
>
> The type of data source often dictates the reliability and comprehensiveness of a dataset. LawBench and LAiW have reorganized traditional datasets to effectively utilize existing data for evaluating LLMs. However, traditional legal tasks were not specifically designed for LLMs, and their limited scope does not fully capture the capabilities of LLMs in the legal domain. Critical areas such as Legal Calculation, Legal Translation, Judicial Analysis Generation, and ethical and moral evaluation are inadequately represented in these datasets.
>
> In contrast, CoLLaM integrates data from existing datasets, real-world exams, and expert annotations. We not only utilize existing datasets but also collect high-quality, real-world data from the National Uniform Legal Profession Qualification Examination and engage legal experts for annotation. This diverse combination of data sources enriches CoLLaM’s tasks and provides a more comprehensive evaluation of LLMs.
>
> **Taxonomy**
>
> A robust taxonomy is essential for organizing and standardizing different tasks within a dataset. A well-structured taxonomy allows for systematic organization, clear expression, and scientific evaluation of various tasks under a unified framework.
>
> LawBench organizes tasks into three levels: Memory, Understanding, and Application. LAiW categorizes tasks into three types: Basic Information Retrieval, Legal Foundation Inference, and Complex Legal Application. While these taxonomies effectively structure tasks within their respective datasets, they miss critical areas in legal domain evaluation, such as ethical reasoning and generative capabilities.
>
> CoLLaM introduces a Legal Cognitive Ability Taxonomy (LCAT), comprising six levels: Memorization, Understanding, Logic Inference, Discrimination, Generation, and Ethics. This taxonomy is designed to reflect the comprehensive capabilities that LLMs need to develop in the legal domain, covering a broad spectrum of legal tasks. LCAT is a distinctive innovation of CoLLaM, providing a more robust and inclusive framework for task organization. This richer taxonomy lays the foundation for a thorough evaluation of LLM capabilities in the legal domain.
>
> **Task Setup**
>
> In terms of task setup and data scale, CoLLaM stands out as the largest and most comprehensive legal dataset in China, encompassing 23 tasks with a total of 13,650 datas.
>
> We also compared the distribution of tasks across CoLLaM, LAiW, and LawBench. CoLLaM includes 11 out of LaiW's 14 tasks, with the exceptions being Case Recognition, Case Understanding, and Legal Consultation. For LawBench, 15 out of its 20 tasks have similar counterparts in CoLLaM. Some tasks in LawBench, such as Fact-based Article Prediction and Scene-based Article Prediction, are combined into a single task in CoLLaM (Article Prediction). The tasks from LawBench not covered by CoLLaM are Dispute Focus Identification, Marital Disputes Identification, Issue Topic Identification, Trigger Word Extraction, and Consultation.
>
> However, CoLLaM introduces 10 unique tasks not found in the other benchmarks, including Legal Rule (1-2), Legal Evolution (1-3), Legal Fact Verification (2-2), Multi-hop Reasoning (3-4), Legal Calculation (3-5), Legal Translation (5-3), Open-ended Question Answering (5-4), Bias and Discrimination (6-1), Morality (6-2), and Privacy (6-3). These additions highlight CoLLaM’s broad scope and its capacity to evaluate a diverse range of legal capabilities.
>
> In addition, CoLLaM enhances standardization by converting tasks into both multiple-choice and generation formats. This approach not only standardizes evaluation methods and metrics but also provides a robust foundation for the future expansion and integration of diverse tasks. By employing these standardized formats, CoLLaM facilitates a more consistent and comprehensive assessment framework, supporting the advancement and adaptability of legal research and applications.

---

> ### Author Response · Authors · 2024-08-17
> **Manual Evaluation of Generation tasks**
>
> In this section, we describe the manual evaluation of four Generation tasks: Summary Generation (5-1), Judicial Analysis Generation (5-2), Legal Translation (5-3), and Open-ended Question Answering (5-4). The following details outline our procedures.
>
> Due to cost constraints, we randomly sampled 50 queries from each task. We collected zero-shot outputs from the top six models in CoLLaM: two closed-source models (GPT-4, ChatGPT), two open-source models (Qwen-14B-Chat, Qwen-7B-Chat), and two legal-specific models (Fuzi-Mingcha, Chatlaw-33B). In total, we gathered 1,200 query-answer pairs. Each task and answer pair was annotated three times, resulting in a total of 3,600 annotations. We paid 0.22 dollars for each annotation task, resulting in a total expenditure of 792 dollars.
>
> Specifically, we employed three legal experts, all of whom have passed the National Uniform Legal Profession Qualification Examination. These experts are from China, including two males and one female. To ensure high-quality annotation, all annotators completed several hours of training to thoroughly understand the evaluation criteria. We assessed their performance using various examples to confirm their competency. Additionally, we provided annotators with the ground truth for the questions to help them grasp the key points of the answers more effectively. Each task and answer pair was annotated three times, and we used the Kappa statistic to measure the consistency and quality of the annotations. The Kappa values for tasks 5-1, 5-2, 5-3, and 5-4 are 0.5888, 0.754, 0.723, and 0.635, respectively, indicating the reliability of our annotations.
>
> We employed a 5-level pointwise scale for annotation. Annotators were provided with a (text, summary) pair or (question, answer) pair and were required to rate the quality on a 5-point scale. We adopted the Likert scale[1] as the 5-level annotation rule. For example, for the statement “The summary text adequately and briefly summarizes the core meaning of the original case” levels 1 ∼ 5 respectively represent the annotator strongly disagrees/disagrees/neutralizes/agrees/strongly agrees with the above statement. Detailed annotation standards are provided in our GitHub https://github.com/CSHaitao/CoLLaM.
>
> The results of the manual annotations are presented in Table 2 of the attachment PDF. We have also included the average BERTScore, BARTScore, and ROUGE-L scores for the six models on this dataset, along with their rankings for reference.
>
> Our findings from the experimental results are as follows:
>
> * GPT-4 performed the best in human evaluations, followed by Qwen-14B. The overall ranking aligns with the model performance ranking presented in Table 3 of the original paper, suggesting that CoLLaM reflects real human evaluations to some extent.
> * For the evaluation of Generation tasks, we reported BERTScore, BARTScore, and ROUGE-L results in the table. We observed that these metrics differ to some extent from human evaluations. For instance, GPT-4, which performed best in human evaluations, ranked 4th according to ROUGE-L.
>
> Then, we calculated Spearman's rank correlation coefficient, Kendall rank correlation coefficient, and Pearson correlation coefficient between the three metrics and human evaluation results. The results are as follows:
>
> | Metrics                                  | BERTScore | BARTScore | Rouge-L |
> | ---------------------------------------- | :-------: | :-------: | :-----: |
> | Spearman's rank  correlation coefficient |   0.829   |    0.6    |  0.257  |
> | Kendall rank correlation coefficient     |   0.733   |   0.467   |   0.2   |
> | Pearson Correlation Coefficient          |   0.984   |   0.969   |  0.792  |
>
> The table data indicates that these metrics may not fully reflect the results of human annotations but can still provide some insight into the general capabilities and trends of the LLMs. Further research is needed to develop more effective automated evaluation methods for assessing the quality of generated text.
>
> Overall, BERTScore shows the highest consistency with human evaluations. Therefore, when cost constraints are a factor, we recommend using BERTScore as an alternative to human evaluations.

---

### Author Rebuttal · Authors · 2024-08-17

# Overview

We would like to express our sincere gratitude to all the reviewers for their thoughtful and constructive feedback. We are pleased to hear that the reviewers recognized the **sound design philosophy** behind constructing CoLLaM [tqv1, NYnk, AqtW], **the reliability of our evaluation results** [tqv1, Nynk, Houv], **the breadth and number of tasks** [tqv1, NYnk, AqtW], and **the richness of the documentation** [tqv1, NYnk].

In response to the reviewers' comments, we have made several revisions and are pleased to report the following updates to the paper:

**1. Comparison with Existing Legal Benchmarks**

We have conducted a more detailed comparison between CoLLaM and previous legal benchmarks. Table 1 in the attached PDF provides a summary comparison of existing datasets. Additionally, we have included a dedicated section in the revised draft that carefully analyzes and discusses the differences in terms of data sources, taxonomy, and task setups.

**2. Additional Experiments on CoLLaM**

We now report a significant number of additional experiments, highlighting three specific improvements:

* **More Nuanced Metrics:** We have provided more nuanced metrics to offer a better and more comprehensive assessment. Specifically, for multiple-choice tasks, we calculated Accuracy and Micro-F1. For generative tasks, we computed three automated metrics: Rouge-L, BERTScore, and BARTScore.
* **Human Evaluation:** We developed a detailed guideline for human evaluation and recruited legal experts to conduct human assessments on the Generation tasks.
*   **Fine-Tuning Experiments with BERT-based Models:** We have supplemented our work with fine-tuning experiments on Chinese-BERT. We found that fine-tuned BERT-based models perform significantly worse on CoLLaM compared to LLMs. We have carefully analyzed the reasons behind these findings, demonstrating that CoLLaM is a challenging benchmark designed for the evaluation of LLMs.

3. **Clarifications on Reviewers' Concerns**

We have addressed and clarified several concerns raised by the reviewers, including:
*   The potential simplification of tasks due to the multiple-choice format.
*   The necessity and importance of the taxonomy design.
*   The focus on the Chinese legal system.
*   Some Details related to data annotation and data representativeness.

We provide the additional experimental results in the attached PDF.

Once again, we greatly appreciate the reviewers' thorough and detailed comments and look forward to addressing any remaining questions or concerns.

---

### Decision · Program_Chairs · 2024-09-26

**Decision:**

Accept (Poster)

**Comment:**

This paper introduces a new largest-so-far, dataset for Chinese legal evaluation with 23 tasks and 13,650 questions. The tasks are categorized using a new taxonomy of legal cognitive abilities that categorizes tasks into six levels: Memorization, Understanding, Logic Inference, Discrimination, Generation, and Ethics. They construct the dataset from three sources - 1) existing datasets (CAIL< JEC-QA, LeCARD etc), (2) National Uniform Legal Profession Qualification Examination, and (3) new expert annotations.

They evaluate with 38 different LLMs -- open, closed and law-specific, under both zero and few-shot settings. Their results show among other things, that while LLMs are good at tasks at the legal understanding and logic inference levels, they fall short at the discrimination, generation and memorization levels. The release the dataset and a leaderboard at https://github.com/CSHaitao/CoLLaM.

+ All reviewers appreciate the rigor and effort put into the dataset creation and annotation, the soundness of the design philosophy, and the thorough documentation of the entire data collection procedure.
+ All reviewers appreciate the thorough and exhaustive set of LLMs evaluated in a variety of settings
+ All reviewers find the paper organization and writing quality to be high, and appreciate the presentation of an adequate number of examples for each task.
+ Overall, reviewers think that the entire dataset construction and evaluation process is quite solid.

- Some reviewers were concerned with using the MCQ format to evaluate models, which might make the task simpler, or may not correlate with LLM generation performance.
- Some reviewers wanted results on Fine-tuned LLMs, as well as more advanced metrics such as BERTScore and BARTScore.
- Reviewers wanted inter annotator scores as well as a detailed comparison with LawBench and other benchmarks.

** The authors have provided “massive” rebuttals to clarify all the above points. The rebuttals can probably be concatenated to form a full second paper. However, reviewers have been satisfied by the rebuttals.  The overall scores indicate a weak accept.